# Structural basis for the disaggregase activity and regulation of Hsp104

**Alexander Heuck[1†], Sonja Schitter-Sollner[1†], Marcin Józef Suskiewicz[1†], Robert Kurzbauer[1], Juliane Kley[1], Alexander Schleiffer[1], Pascaline Rombaut[2], Franz Herzog[2], Tim Clausen[1]***

[1]Research Institute of Molecular Pathology, Vienna Biocenter, Vienna, Austria; [2]Gene Center and Department of Biochemistry, Ludwig-Maximilians University, Munich, Germany

**Abstract** The Hsp104 disaggregase is a two-ring ATPase machine that rescues various forms of non-native proteins including the highly resistant amyloid fibers. The structural-mechanistic underpinnings of how the recovery of toxic protein aggregates is promoted and how this potent unfolding activity is prevented from doing collateral damage to cellular proteins are not well understood. Here, we present structural and biochemical data revealing the organization of Hsp104 from *Chaetomium thermophilum* at 3.7 Å resolution. We show that the coiled-coil domains encircling the disaggregase constitute a 'restraint mask' that sterically controls the mobility and thus the unfolding activity of the ATPase modules. In addition, we identify a mechanical linkage that coordinates the activity of the two ATPase rings and accounts for the high unfolding potential of Hsp104. Based on these findings, we propose a general model for how Hsp104 and related chaperones operate and are kept under control until recruited to appropriate substrates.

*For correspondence: clausen@
imp.univie.ac.at

†These authors contributed
equally to this work

Competing interests: The
authors declare that no
competing interests exist.

Reviewing editor: Manajit
Hayer-Hartl, Max Planck Institute
of Biochemistry, Germany

## Introduction

Proteins, the most intricate of biological macromolecules, are inherently prone to misfolding and aggregation. These processes, aggravated by mutations, translation errors, aging, and physico-chemical stresses (*Tyedmers et al., 2010*), can be toxic and are linked to severe human diseases (*Horwich, 2002*). As part of the protective response, cells of all organisms produce chaperones of the AAA (ATPases Associated with a variety of cellular Activities) family that utilize the energy from ATP hydrolysis to remodel non-native proteins. The AAA chaperone machines can associate with cage-forming proteases, such as the 20S proteasome, yielding bipartite proteolytic complexes, or, alternatively, team up with partner chaperones to solubilize aggregated proteins and promote their refolding (*Sauer et al., 2004*).

HSP100 unfoldases, a subclass of the AAA chaperones found in yeast and bacteria as well as in the mitochondria and chloroplasts of higher eukaryotes, employ a powerful mechanism to recover functional proteins from aggregates. Upon forming a hexameric ring, they unravel polypeptides by threading them through a narrow central pore (*Weber-Ban et al., 1999*; *Weibezahn et al., 2004*; *Hinnerwisch et al., 2005*). Substrate stretching and unfolding is mediated by ATP-driven power strokes (*Maillard et al., 2011*), which result from the movement of rigid ATPase bodies composed of the large (L) subdomain of one protomer and the small (S) subdomain of the next (*Glynn et al., 2009*; *Wang et al., 2001*). Coordination of adjacent L/S* modules (the asterisk denotes the neigh-boring subunit) relies on a special active site organization, as each nucleotide binding site is formed by residues of the L-, S*- and L*-subdomains at the subunit interface. In terms of substrates, the particularly powerful HSP100 unfoldases have the remarkable ability to disentangle protein aggregates (*Parsell et al., 1994*), which, owing to their inert, scrambled, and water-insoluble character,

represent the most challenging target for the protein quality control system. Although HSP100 chaperones play a crucial function in removing these potentially dangerous aggregates, the molecular details of their robust cleaning activity have remained unclear. It has been postulated that the disaggregation activity of HSP100 machines relies on the presence of two AAA rings, most likely to provide a strong, 2-handed grip for remodeling protein substrates (*Doyle et al., 2007a*; *Olivares et al., 2014*; *Hinnerwisch et al., 2005*). However, the structural basis underlying the functional coupling of the two ATPase rings has not been determined. To address this mechanism, we performed a structure-function analysis of the fungal Hsp104, a disaggregase that was originally identified as a critical factor for survival under extreme stress conditions (*Sanchez and Lindquist, 1990*; *Sanchez et al., 1992*) and, later, for prion propagation (*Chernoff et al., 1995*). Indeed, Hsp104 can team up with Hsp70 to establish one of the most potent disaggregase machineries in nature, being able to unravel even the particularly resistant amyloid fibers (*Shorter and Lindquist, 2004*; *Inoue et al., 2004*; *Krzewska and Melki, 2006*). To avoid damage to native proteins, the high unfolding potential of Hsp104 and related disaggregases needs to be carefully regulated. This control is mediated by an inserted coiled-coil domain (M-domain, MD), which assembles a molecular belt encircling the hexameric particle and keeping the enzyme in its latent state (*Carroni et al., 2014*). Binding of the Hsp70 chaperone to the MD activates Hsp104 and targets it towards protein aggregates (*Lee et al., 2013*; *Seyffer et al., 2012*; *Rosenzweig et al., 2013*; *Oguchi et al., 2012*; *Haslberger et al., 2007*; *Miot et al., 2011*; *Sielaff and Tsai, 2010*). Again, despite the wealth of genetic and biochemical data and the availability of structural information (*Carroni et al., 2014*; *Lee et al., 2007*, *2003*, *2010*; *Yokom et al., 2016*), the molecular mechanism of how the MD regulates the disaggregase machinery could not be resolved so far.

Here, we present the crystal structure of Hsp104 from *Chaetomium thermophilum* that – although forming a helical filament – reveals important mechanistic features. First, we identify the long-sought mechanical link coupling the two AAA rings of HSP100 chaperones and, second, we delineate structural details underlying the regulatory role of the MD. Jointly, these two elements make Hsp104 a very potent yet highly tunable protein disaggregase. As will be discussed, the uncovered mechanistic features represent novel concepts that might be generally applicable to AAA mechanoenzymes implicated in protein quality control and beyond.

## Results

### Crystal structure of the Hsp104 subunit

To study the molecular details of the HSP100 machinery, we performed a biochemical and structural analysis of the Hsp104 disaggregase from *Chaetomium thermophilum* (CtHsp104), which exhibits similar ATPase and protein remodeling activities to those of the well-characterized *Saccharomyces cerevisiae* ortholog (ScHsp104) (*Figure 1A*). The crystal structure of the double Walker-B mutant in complex with ADP was determined at 3.7 Å resolution (*Table 1*). Despite medium resolution, the final electron density map was of excellent quality revealing the overall side-chain conformation of most functionally important residues (*Figure 1—figure supplement 1*).

The Hsp104 disaggregase is composed of tandemly linked structural modules that build up the N-terminal domain (NTD), the two ATPase engines (AAA1, AAA2), and the regulatory MD extension (*Figure 1B*). Although the obtained crystals are formed by the full-length protein (data not shown), the NTD was not defined by electron density, likely due to high flexibility. Such *en-bloc* mobility was also reported in previous EM studies of related HSP100 enzymes (*Ishikawa et al., 2004*; *Effantin et al., 2010*; *Carroni et al., 2014*; *Lee et al., 2003*, *2007*) and seems to be an inherent property of the domain, which is connected to the rest of the molecule by a long linker. The AAA1 and AAA2 domains consist of a large (L) RecA-like α/β and a small (S) α-helical ATPase subdomain. Like in the crystal structure of the closely related ClpB disaggregase (*Lee et al., 2003*) that operates by the same basic mechanism as Hsp104 (*Kummer et al., 2016*), the large subdomains AAA1L and AAA2L are organized around a five-stranded parallel β-sheet that is flanked by three to four helices on either side. The small C-terminal subdomains AAA1S and AAA2S have a four-helix bundle at their core, with one helix replaced by a three-stranded β-sheet in AAA2S. The active sites, located at the AAA subdomain interface, are defined by co-crystallized ADP molecules, which were clearly visible in the electron density map (*Figure 1—figure supplement 2*). In AAA1, ADP is accommodated in a

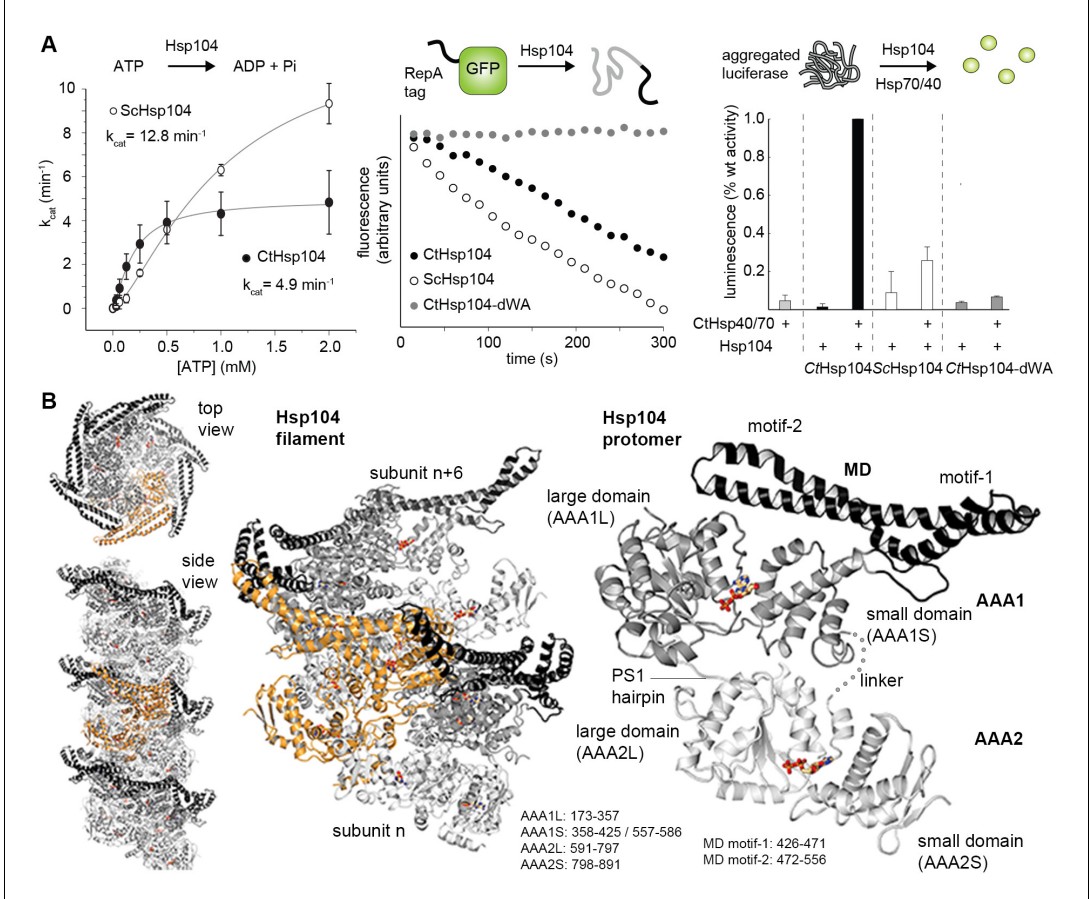

**Figure 1.** Crystal structure of Hsp104. (**A**) Overview of ATPase, GFP unfolding, and luciferase disaggregation assays used to test the functionality of the Hsp104 protein from *C. thermophilum* (CtHsp104) and its homologue from yeast (ScHsp104). The catalytic inactive double Walker A (WA; K229A/K640A) mutant was used as negative control. For the unfolding and disaggregase assay, the relative activities to the CtHsp104 wild-type protein are shown, with error bars representing standard deviations of 3 independent experiments. Unfoldase activity was measured using a non-physiological ATP/ATPgS mixture required by the assay setup. (**B**) Ribbon presentation of the crystallized Hsp104 filament (color coded according to domains, with one whole subunit depicted in orange) and of the constituting protomer. ADP molecules are shown in stick representation. See also *Figure 1—figure supplements 1* and *2*.

The following figure supplements are available for figure 1:

**Figure supplement 1.** Crystal structure of Hsp104 from *C. thermophilum*.

**Figure supplement 2.** Active sites of AAA1 and AAA2.

pocket formed by the general Walker A (Lys229, Thr230) and Walker B (Asp295, Glu296) motifs as well as by AAA-specific (sensor-1: Thr330, sensor-2: Arg402) functional groups (*Hanson and White-heart, 2005*; *Mogk et al., 2003*; *Hattendorf and Lindquist, 2002b*). In addition, the neighboring subunit contributes one α-helix, the so-called Second Region of Homology (SRH) motif, to the nucle-otide-binding pocket. From here, Arg349* protrudes into the active site and interacts with the Pα and Pβ of the bound ADP (*Figure 1—figure supplement 2*). Consistent with its position close to the nucleotide phosphates, Arg349* has been previously described as the arginine finger critical for ATP binding and hydrolysis, and inter-subunit communication (*Mogk et al., 2003*). In AAA2, the active site is formed by the corresponding motifs (Walker A: Lys640, Thr641; Walker B: Asp706, Glu707; sensor-1: Asn748; sensor-2: Arg849 and SRH*: Arg788 (*Biter et al., 2012b*; *Mogk et al., 2003*; *Hattendorf and Lindquist, 2002b*, *2002a*)), though participating side-chains are less defined by electron density than in AAA1. In addition to exhibiting a lower overall flexibility, AAA1 is

**Table 1.** Data collection and refinement statistics.

| | 5d4w |
|---|---|
| *Data collection* | |
| Space group | P2$_1$ |
| Cell dimensions | |
| a, b, c (Å) | 144.9, 93.2, 144.4 |
| α, β, γ (°) | 90, 119.7, 90 |
| Resolution (Å) | 47.2–3.7 (3.8–3.7)[1] |
| $R_{merge}$ | 0.192 (1.624) |
| $R_{pim}$ | 0.130 (1.121) |
| I/σI | 5.3 (0.8) |
| Completeness (%) | 98.3 (97.9) |
| Redundancy | 3.1 (2.9) |
| *Refinement*[2] | |
| Resolution (Å) | 47.6–3.7 |
| No. reflections | 35560 |
| $R_{work}$/$R_{free}$ | 0.237/0.277 |
| No. atoms (all) | 16272 |
| Protein | 16110 |
| ADP | 162 |
| B-factors (all) | 210.7 |
| Protein | 210.9 |
| ADP | 188.6 |
| R.m.s deviations | |
| Bond lengths (Å) | 0.006 |
| Bond angles (°) | 1.082 |

[1]Highest resolution shell is shown in parenthesis.

[2]One crystal was used for measurement and the stereochemistry of the model was validated with Molprobity (**Chen et al., 2010**).

distinguished from AAA2 by a prominent additional structure, the MD, which is an extension of AAA1S. The MD motif comprises an elongated helix that pairs with its flanking helices via Leu-zipper interactions. The C-terminal blade (motif-2) of the resultant propeller-like structure binds to the AAA1 body via numerous polar contacts, while the N-terminal blade (motif-1) protrudes away from the subunit to interact with a neighboring Hsp104 protomer.

## ATPase rigid bodies are maintained in the crystallized Hsp104 filament

In the crystal, CtHsp104 subunits are arranged in a helical 6$_1$ filament (*Figure 1B*) rather than forming a defined hexameric particle as observed in the recent cryo-electron microscopy structure of ScHsp104 (*Yokom et al., 2016*) or the crystal structure of a related HSP100 unfoldase, ClpC (*Wang et al., 2011*). It is interesting to note that while the ClpC hexamer is planar, the ScHsp104 hexamer shows a helical, staircase-like arrangement (*Figure 2—figure supplement 1*) with a smaller pitch but the same handedness as our crystallized filament. This suggests that CtHsp104 crystallization might have built upon an intrinsic tendency of Hsp104, and perhaps all HSP100 proteins, to switch between planar and helical conformations. The uncanonical AAA1-AAA2 interface between the first and the sixth protomer that closes the ring in the case of ScHsp104 is absent in our CtHsp104 crystal, allowing an infinite spiral to form. Importantly, however, the crystallized filament seems to reflect mechanistic properties of the HSP100 hexamer, as the architecture of the basic building blocks of AAA unfoldases, the so-called rigid bodies, is comparable to that in both ClpC and ScHsp104 hexameric structures. As seen in previous crystal structures of hexameric unfoldases, the AAA rigid body comprises a mixed L/S* module formed between adjacent subunits (*Glynn et al., 2009*; *Wang et al., 2001*). In fact, structural comparison shows that the AAA1L/AAA1S* and AAA2L/AAA2S* rigid bodies of the Hsp104 filament are very similar to those of the hexameric ScHsp104 and ClpC (*Figure 2* and *Figure 2—figure supplement 2*). A central component tethering the L and S* subdomains is the sensor-2* helix, the helix following the sensor-2 residue (*Hanson and Whiteheart, 2005*), that accounts for about 75% of the inter-subunit interface (750 Å$^2$ in total). Oriented by specific polar contacts, the sensor-2* helix protrudes towards the active site, with its N-terminal end located in a close distance from the arginine fingers that protrude from the SRH helix (*Figure 2B* and *Figure 1—figure supplement 2*). In concert, the properly arranged sensor-2* and SRH residues can perform their nucleotide-sensing task and coordinate the interplay of ATPase rigid bodies (*Glynn et al., 2009*; *Wang et al., 2001*). It is important to note that, in contrast to the L/S* organization of Hsp104, helical crystal structures of the bacterial homologue ClpB (*Carroni et al., 2014*; *Lee et al., 2003*) do not reveal the functional L/S* rigid bodies (*Figure 2* and *Figure 2—figure supplement 2*). Here, the sensor-2 helix does not pair with the neighboring subunit and, consequently, the composite sensor-2*/SRH motif is disrupted. Moreover, the two subdomains are connected by a markedly reduced interface of 100–400 Å$^2$, which is too small to support *en-bloc*

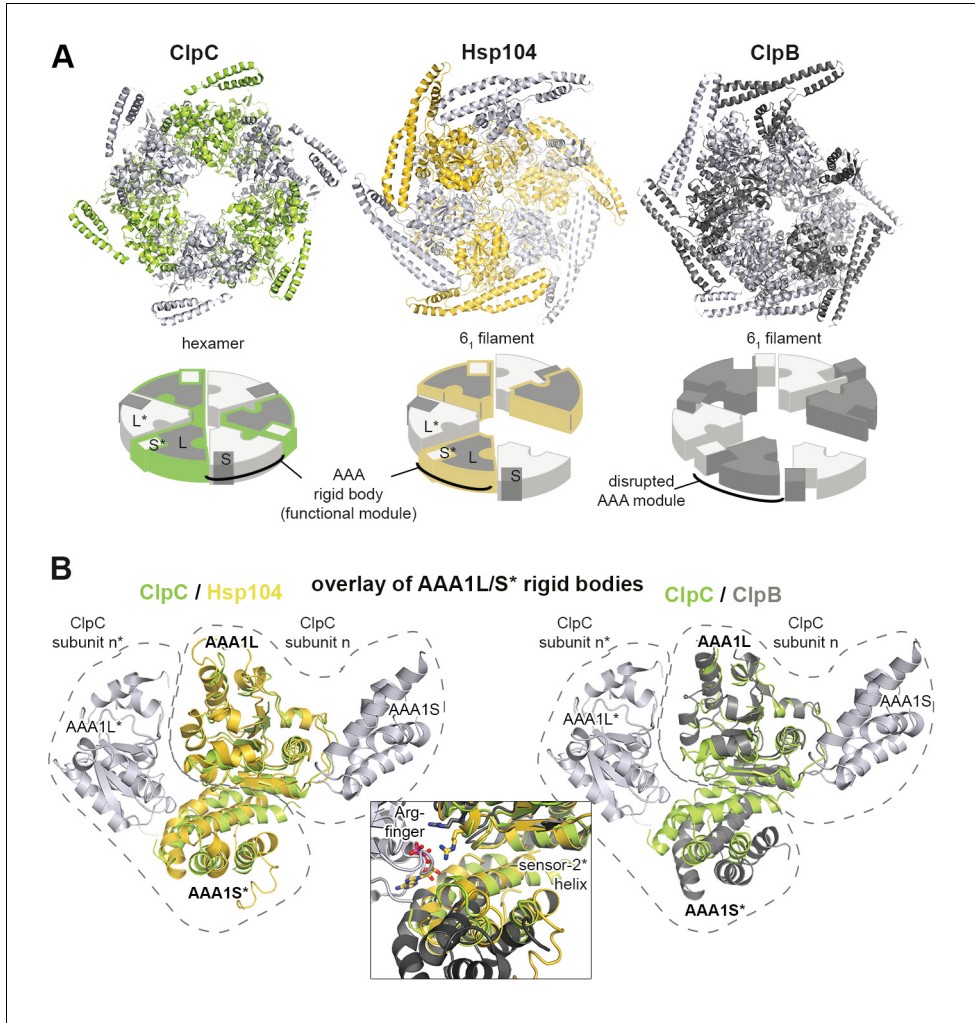

**Figure 2.** Functional ATPase modules are retained in the Hsp104 filament. (**A**) Ribbon presentation of ClpC (PDB 3pxg), Hsp104, and ClpB (PDB 4ciu) oligomers highlighting the L/S* rigid bodies (colored). While ClpC was crystallized as a hexamer, Hsp104 and ClpB were crystallized in a helical filament with a six-fold screw axis. The cartoons below depict six L/S* modules (framed) that are formed between adjacent subunits (distinct grey tones). (**B**) Ribbon presentation of adjacent AAA1 domains of the hexameric ClpC that jointly compose one L/S* rigid body (green). Superimposed are the L/S* modules of the Hsp104 (yellow, left panel) and ClpB (dark grey, right panel) filament. The zoomed-in window emphasizes the distinct orientation of the sensor-2 helix, the central element in linking L and S* sub-domains. ADP and the SRH* arginine finger are shown in stick representation to mark the position of the active site. See also *Figure 2—figure supplements 1,2* and *3*.

The following figure supplements are available for figure 2:

**Figure supplement 1.** Comparison between planar and helical Hsp100 conformations.

**Figure supplement 2.** Functional ATPase modules are retained in the Hsp104 filament.

**Figure supplement 3.** Sequence conservation of HSP100 disaggregases.

movement of L and S*. In conclusion, the structural alignments suggest that the L/S* rigid bodies of the Hsp104 filament are similarly organized to those of the native hexameric enzyme. This notion is further corroborated by the observation that residues mediating the Hsp104 L/S* contacts are highly conserved (*Figure 2—figure supplement 3*) and match the hydrogen-exchange data of the related ClpB disaggregase (*Oguchi et al., 2012*). Together, these points suggest that the Hsp104 structure

provides a molecular model of unprecedented resolution to study mechanistic aspects of HSP100 function. In fact, structure-guided experiments allowed us to biochemically determine the regulatory role of the MD and to discover a mechanical link establishing the potent two-engine disaggregase motor of Hsp104.

## The Hsp104 structure reveals the regulatory contacts of the MD

Recent studies highlight the central role of the MD in regulating HSP100 disaggregases, as the domain is critical for AAA1-AAA2 communication, Hsp70 binding, and keeping the enzyme inactive in the absence of cognate substrates (*Cashikar et al., 2002*; *Lee et al., 2013*; *Oguchi et al., 2012*; *Seyffer et al., 2012*; *Haslberger et al., 2007*; *Sielaff and Tsai, 2010*; *Miot et al., 2011*). As described, the MD assembles a coiled-coil propeller that consists of motif-1 (residues 426–471) and motif-2 (residues 472–556). The adjacent propellers can interact in a head-to-tail fashion, whereby the tip of motif-1 contacts that of motif-2*, yielding a ring-like scaffold encircling the six AAA1 domains (*Carroni et al., 2014*). Binding of Hsp70 to the tip of motif-2 turns Hsp104 activity on, presumably by abrogating the MD-MD* interaction. Of note, site-specific mutations influencing the MD contacts have a similar regulatory effect (*Supplementary file 1*). They can either disrupt the coiled-coil contacts, yielding a toxic, hyperactive Hsp104 variant that is constitutively 'on' (*Lee et al., 2005*; *Lipińska et al., 2013*; *Schirmer et al., 2004*; *Oguchi et al., 2012*) or stabilize the MD-MD* interaction, yielding a repressed form that is strongly inhibited (*Carroni et al., 2014*). Although previous studies revealed an influence of the MD on the overall AAA1 domain conformation (*Oguchi et al., 2012*; *Lee et al., 2007*), an effect relying on a number of polar interactions between the two domains (*Lipińska et al., 2013*), the precise molecular mechanism of regulating HSP100 disaggregases has not been determined.

The present crystal structure provides a detailed picture of the resting state of Hsp104, shedding light on the regulatory role of the MD. In particular, intramolecular contacts between the MD and the ATPase core are for the first time delineated with high precision. Accordingly, three main interfaces can be distinguished (*Figure 3A*). First, an extensive network of polar interactions connects the central portion of the MD propeller to the AAA1 small subdomain of one ATPase module. In parallel, the MD binds to the neighboring ATPase module by docking the tip of motif-2, particularly Arg509, into a two-helix cleft on the edge of the AAA1 large subdomain. The two elements engaged by the MD, AAA1S and AAA1L, belong to the same subunit but to different ATPase modules. This tethering of adjacent ATPase modules is of utmost importance for the regulation of Hsp104 (see below). Finally, several van-der-Waals contacts and hydrogen bonds connect the tips of motif-1 and motif-2* of two adjacent coiled-coils, yielding a continuous MD ring that wraps around the AAA1 core. The functional relevance of this interaction network is underscored by the fact that every residue in the list of hyperactive mutations overlaps with one of the specific MD-AAA1 or MD-MD* contacts observed in the crystal structure (*Figure 3A* and *Supplementary file 1*). To further test the validity of the described network, we mutated two additional residues at the MD interfaces (D247A and A446V) and assessed their enzymatic activity (*Figure 3B*). As in previous analyses (*Biter et al., 2012b*; *Jackrel et al., 2014*), the stimulatory effect on the unfolding and disaggregase activities varied to some extent, possibly because these functions depend on the interaction with substrates and co-chaperones that may be affected by mutations in the MD. Most importantly, however, the predicted hyperactive mutants displayed an elevated basal ATPase activity, confirming that the contacts seen in the crystal structure are critical to regulate Hsp104.

## The MD belt controls the mobility and thus activity of AAA1 modules

Having revealed the precise architecture of the inter-domain interface, we next studied how the intricate interaction network linking the MD and AAA1 domains could repress Hsp104 activity. In this regard, it should be noted that for AAA unfoldases to work, the ATPase modules need to move against each other in order to mechanically unfold client proteins (*Olivares et al., 2016*). As seen in the proteasome, the ATP-driven conformational changes comprise pronounced *en-bloc* movements of the L/S* modules of about 15 Å (*Śledź et al., 2013*), which would be hindered by an enclosing structural scaffold. To estimate the corresponding ATPase rearrangements in Hsp104, we generated a model of the disaggregase hexamer using the ClpC hexamer (*Wang et al., 2011*) as a template (*Figure 4A*). Superposition of the proteasomal ATPase core subunits Rpt$_{1-6}$ on the modeled Hsp104

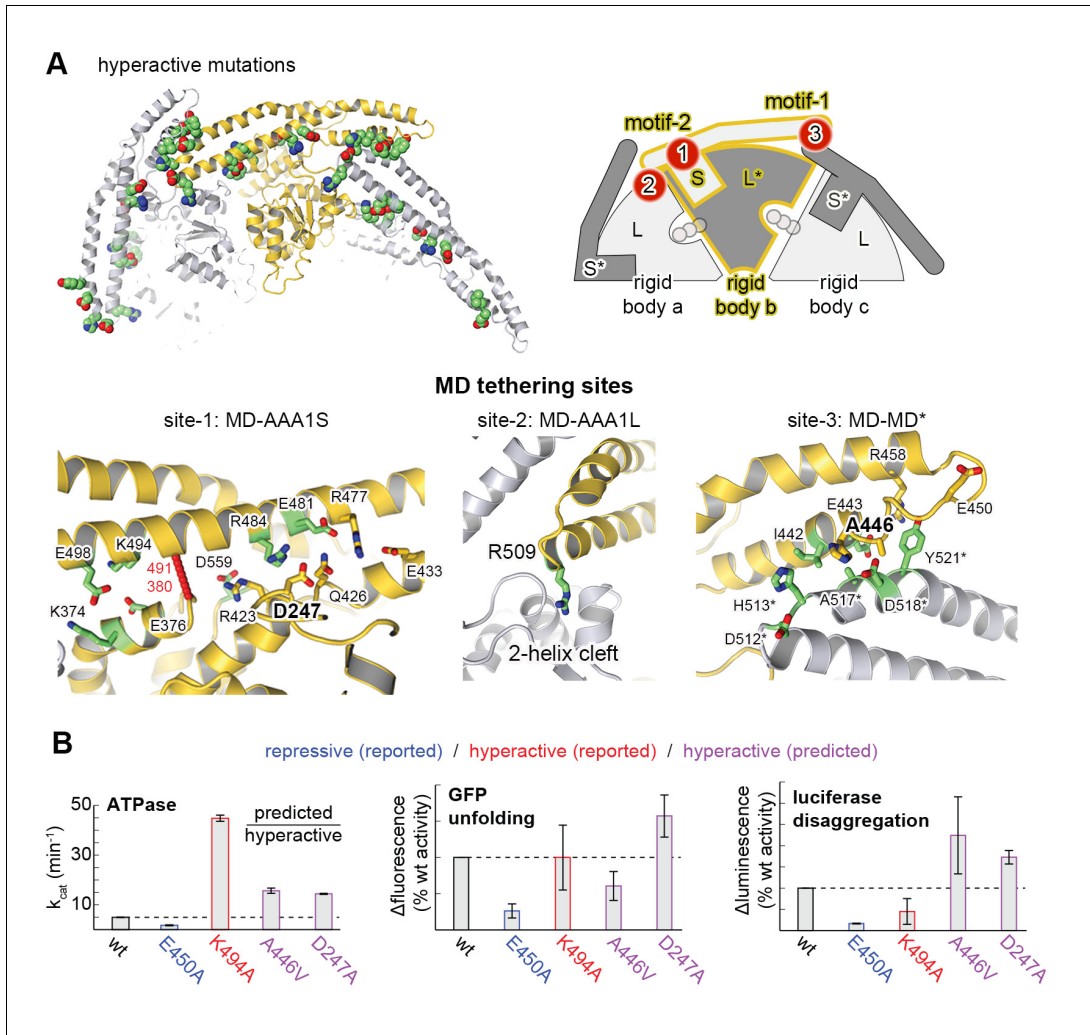

**Figure 3.** Structural organization of the MD-AAA1 interface. (**A**) Ribbon presentation of Hsp104 showing the clustering of hyperactive mutations (green, see **Supplementary file 1**) along the MD-AAA1 and MD-MD* interfaces. The cartoon illustrates the localization of the three major MD tethering sites, which are also shown in structural detail. The 380–491 pair used in cross-linking studies connects motif-2 of the MD with AAA1 (red line). (**B**) Comparison of mutants predicted to be hyperactive (A446V and D247A should destabilize the MD-AAA1-MD* interface) with reported hyperactive (red, K494A) and repressed (blue, E450A) mutants. Error bars indicate standard deviations.

hexamer suggests that the MD scaffold may sterically impede conformational rearrangements of the ATPase rigid bodies (**Figure 4B**). Accordingly, the MD has a potential to establish a topological belt, a 'restraint mask', limiting movement and activity of the entrapped AAA1 domains.

To directly monitor the mobility of the engaged ATPase modules upon opening and closing of the MD ring, we performed a cross-linking coupled mass spectrometry (XL-MS) experiment. As reported recently, the number of distinct Lys-Lys cross-links determined in a comparative XL-MS analysis is useful to estimate protein dynamics in a semi-quantitative manner (**Scorsato et al., 2016**; **Walzthoeni et al., 2015**). Accordingly, we reasoned that a structurally flexible Hsp104 particle that comprises a mixture of different conformations should yield a more complex cross-linking pattern than an Hsp104 variant that is closely embraced by a MD ring. For the cross-linking reaction, we used bis-sulfosuccinimidyl suberate (BS³), which connects lysine side chains located at a distance of 10–30 Å. To account for the distinct cross-linking efficiencies, we individually adjusted the amounts of BS³ for each of the analyzed Hsp104 species (**Figure 4—figure supplement 1**). According to the

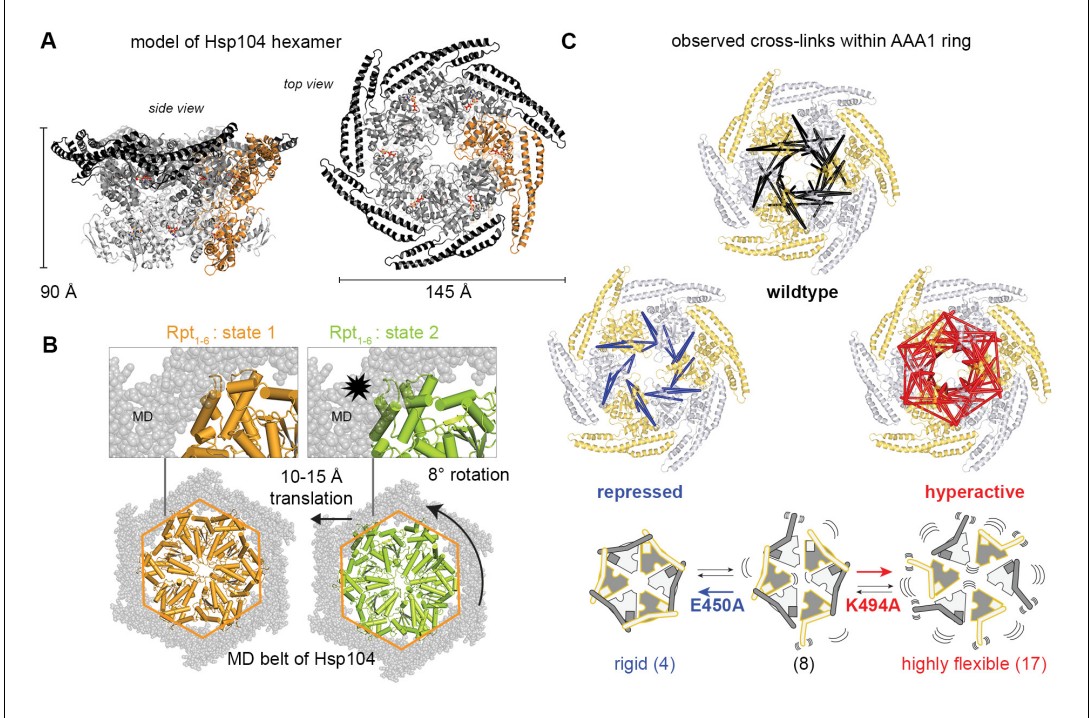

**Figure 4.** Inhibitory function of the MD. (**A**) Ribbon presentation of a modeled Hsp104 hexamer that was constructed from the L/S* rigid body of the crystallized filament. The dimensions of the Hsp104 hexamer are indicated and subunits are colored as in **Figure 1B**). (**B**) Ribbon representation of the proteasomal Rpt$_{1-6}$ present in two alternative conformations (state-1, PDB 4cr2, orange, and state-2, PDB 4cr4, green (**Unverdorben et al., 2014**)). The two AAA rings are shown together with the MD belt (grey surface) of the superimposed Hsp104 hexamer. Conformational differences between the Rpt$_{1-6}$ ATPase modules are indicated. The asterisk highlights a hypothetical clash with the MD belt. (**C**) Ribbon presentation of the modeled Hsp104 hexamer highlighting the L/S* rigid bodies (colored differently). The different numbers of identified Lys-Lys cross-links are represented by intermolecular connections, linking neighboring rigid bodies (wt in black, E450A in blue, K494A in red). The proposed effects of activating and repressing mutations on the dynamics of the Hsp104 hexamer are schematically indicated. See also **Figure 4—figure supplements 1** and **2**.

The following source data and figure supplements are available for figure 4:

**Source data 1.** Cross-linked Lys pairs observed by XL-MS.
**Figure supplement 1.** Analysis of BS$^3$ cross-linking efficiency.
**Figure supplement 2.** Distribution of identified XL-MS cross-links.

restraint-mask model, the mobility of ATPase units should be increased in the hyperactive (K494A) mutant, resulting in a greater number of distinct cross-linked Lys-pairs. In contrast, the repressive (E450A) mutant is expected to be less dynamic, allowing only few distinct Lys-pairs to become cross-linked. Consistent with this prediction, the recorded MS data show clear differences in the number of observed Lys-pairs (**Figure 4—source data 1** and **Figure 4—figure supplement 2**). The most drastic variation was seen within the AAA1 ring, for which we identified four distinct cross-linked Lys-pairs in the repressive, eight in the wild-type, and 17 in the hyperactive Hsp104. Compared to AAA1, differences in the cross-linking pattern within the MD and AAA2 were less pronounced. Mapping the XL-MS data onto the structure of the AAA1 domains immediately illustrates the increased dynamicity of the released ATPase modules in the hyperactive mutant, whereas the repressive mutation seems to freeze the Hsp104 enzyme (**Figure 4C**). In conclusion, the comparative XL-MS analysis supports the restraint-mask model, which predicts that a closed MD belt immobilizes the entrapped AAA1 ATPase modules, thereby keeping the Hsp104 disaggregase in its resting state. Once the MD contacts are broken, the ATPase modules are free to move against each other, as seen for the hyperactive mutant, to remodel engaged client proteins.

## Motif-2 of the MD can tether adjacent AAA1 modules to immobilize them

To further validate the relevance of the MD in sterically controlling Hsp104, we developed a specific cysteine cross-linking approach. From several Cys-Cys pairs inserted in a Cys-free mutant of Hsp104, only a single Cys-Cys couple could efficiently cross-link AAA1 and MD without undergoing side reactions (*Figure 5—figure supplement 1*). Using this G380C-Q491C mutant, it was possible to covalently link motif-2 of the MD to the AAA1S subdomain thereby tethering neighboring ATPase bodies. To estimate the effects of 'tight' and 'loose' MD belts, we carried out Cys-Cys and Cys-bis-maleimidoethane-Cys (BMOE) cross-linking, respectively, and compared the activities to those under no-cross-linking conditions. GFP unfolding and luciferase disaggregation assays revealed that the S-S bond formation upon Cys-Cys cross-linking yielded a fastened MD belt that almost completely abolished activity. In strong contrast, introducing the 8-Å-long BMOE cross-linker at the same position restores flexibility of the MD and supports high unfolding and disaggregation capability (*Figure 5A*). Of note, the introduced cysteine residues at position 380 and 491 influenced by themselves the activity of Hsp104, as predicted by the crystal structure. Whereas, under reducing

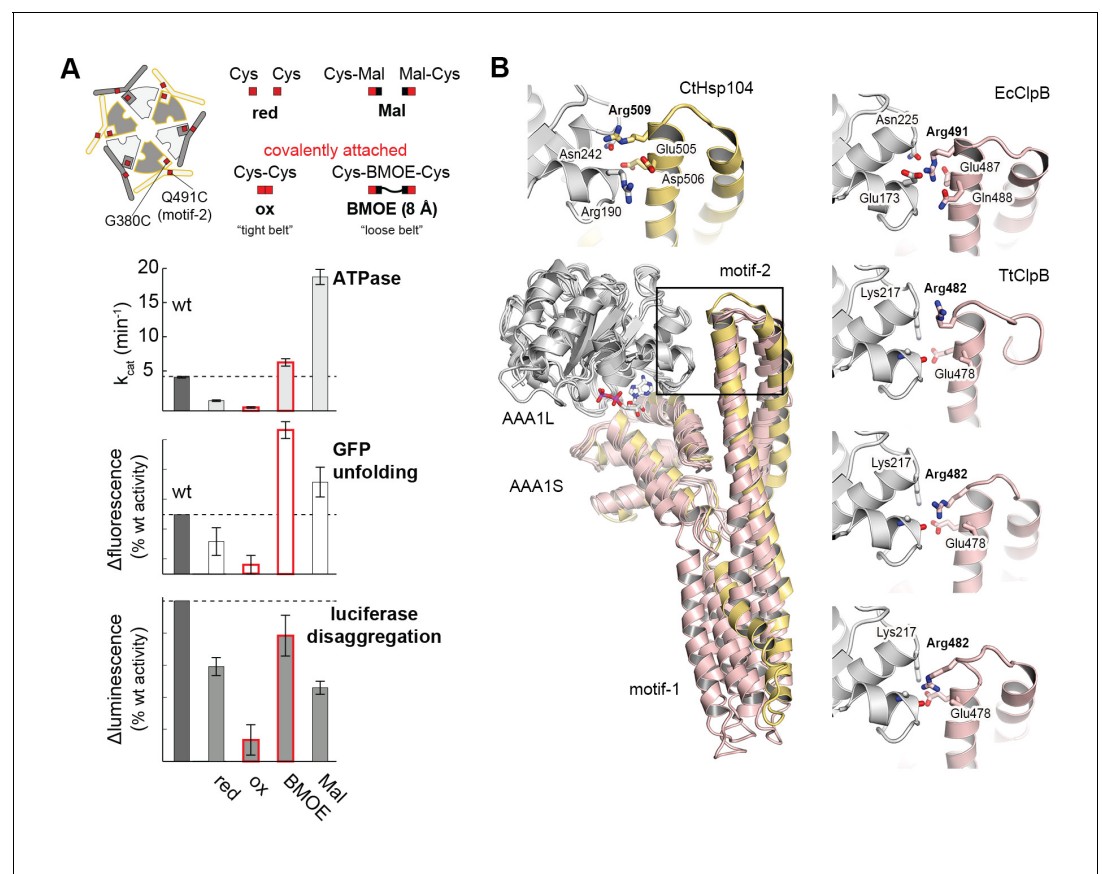

**Figure 5.** Regulatory role of motif-2. (**A**) As schematically shown, the G380C/Q491C mutant allows to covalently connect MD and AAA1 by a 'tight' (Cys-Cys linkage) or a 'loose' belt (Cys-BMOE-Cys). ATPase, unfolding and disaggregase activity of the G380C/Q491C mutant under reducing (red) and oxidizing conditions (ox) and upon reaction with the chemical cross-linker (BMOE) or free maleimide (Mal) are shown in comparison to the wild-type enzyme. Error bars represent standard deviations. See also *Figure 5—figure supplement 1*. (**B**) Interaction between MD and AAA1 domain, observed in Hsp104 and ClpB crystal structures (EcClpB PDB 4ciu, TtClpB PDB 1qvr). As shown in the superposition (Hsp104 MD in yellow; ClpB MD conformations in red) and the zoomed-in windows, motif-2 binds via a conserved arginine to the same two-helix cleft in AAA1L. Adjacent polar contacts should stabilize this interaction.

The following figure supplement is available for figure 5:

**Figure supplement 1.** Identification of a Cys-Cys pair to covalently fix the MD belt.

conditions, the adjacent Cys residues can form a hydrogen bond that further stabilizes the repressed low-activity state, insertion of the bulky BMOE cross-linker (or its functional head, maleimide) seems to physically separate the two Cys residues thereby activating Hsp104 (*Figure 5A*). More importantly, the two cross-linked particles Hsp104(Cys-Cys) and Hsp104(Cys-BMOE-Cys) exhibit distinct activities. While both cross-linkers covalently link and thus physically restrain adjacent ATPase units, only the BMOE compound provides the conformational freedom required for Hsp104 function. In conclusion, the cross-linking data support the restrain-mask model showing that AAA1 domains engaged by a covalently-linked but loosened MD belt can still reorient and cooperate with each other.

The inactivating effect of the Cys380-Cys491 cross-link also highlights the importance of motif-2 as the main inhibitory element that immobilizes two adjacent AAA1 ATPase modules. Structural comparison with ClpB further emphasizes the inhibitory role of motif-2. Superposition of different ClpB structures on Hsp104 points to a pronounced flexibility of motif-1, whereas motif-2 is always similarly oriented (*Figure 5B*). In all structures, the tip of motif-2 is bound to a neighboring AAA1 module, mainly by placing a conserved arginine residue in the described two-helix cleft formed at the AAA1-MD interface (*Figure 3A*). Notably, the respective arginine residue (Arg509) has been identified in Hsp104 as an important functional group giving rise to hyperactive mutations (*Biter et al., 2012a*; *Wendler et al., 2007*).

Taken together, the Cys-Cys cross-linking experiments demonstrate that the MD establishes a topological belt that sterically controls the activity of the Hsp104 disaggregase. The cross-linking data also delineate the key feature of the MD safety belt: By simultaneously binding to two AAA1 rigid bodies, motif-2 is capable of reducing the mobility and thus the activity of engaged ATPase modules. Given the structural conservation of these contacts, we presume that the proposed restrain-mask model is generally relevant for HSP100 disaggregases.

## The PS1-hairpin of AAA2 synchronizes the two AAA rings of Hsp104

The AAA2 domain contains a characteristic β-hairpin that precedes strand β4 and is referred to as the pre-sensor-1 (PS1) motif (*Erzberger and Berger, 2006*). Strikingly, in the crystal structure of Hsp104, the PS1-hairpin protrudes from the AAA2 into the AAA1* ring, where it is accommodated in a deep pocket close to the ATP binding site of a neighboring subunit (*Figure 6A*). To test whether the PS1 motif could adopt a similar conformation in the hexameric particle, we aligned the functional L/S* unit (AAA1L/S*-AAA2L/S*) of the CtHsp104 filament onto the ClpC and ScHsp104 hexamers. Except a few steric clashes with the flexible MD (clashes that could be avoided by a minor reorientation of the coiled-coil domain), the superposed rigid bodies of the ScHsp104 filament, including the PS1 motif, fit remarkably well to the rigid bodies of the planar and helical hexamers (*Figure 6—figure supplement 1*). These structural alignments suggest that the PS1 motif is well positioned to functionally link the AAA1 and AAA2 rings in the various HSP100 oligomers. According to these data, we hypothesize that the Hsp104 disaggregase can switch between planar and helical conformations while maintaining the integrity of the L/S* rigid bodies to ensure intra- and inter-ring cooperativity during the ATPase-driven power strokes.

At the tip of the PS1-haripin, the side chains of Gln732 and Arg734 form specific interactions with Asp406* and Asp410* located on the sensor-2 helix. These contacts, which physically link the PS1-motif of AAA2 to the AAA1S* sub-domain, are in perfect agreement with surprising results of subunit mixing experiments, in which AAA1 exerted an allosteric effect on AAA2 in trans (*Franzmann et al., 2011*). Given its AAA2-AAA1* bridging position and the high conservation of interface residues (*Supplementary file 2*), we explored whether the PS1-hairpin could be the long-sought mechanical link of the two AAA rings in HSP100 chaperones. Upon deleting the PS1-hairpin, we could measure only a minor effect on the overall ATPase activity. Despite leaving the ATPase engines largely untouched, the PS1 deletion, however, fully blocked the unfoldase and disaggregase activities (*Figure 6B*). This remarkable decoupling effect, which has not been reported before for any other HSP100 unfoldase, highlights the importance of the PS1-hairpin for linking the two ATPase engines, a linkage that is essential to achieve full unfolding and disaggregation activity. Notably, deleting the PS1 hairpin in the yeast Hsp104 had a similar effect, suggesting that the uncovered coupling is generally relevant for HSP100 disaggregases (*Figure 6—figure supplement 2*). To show that the observed effects are not due to putative gross structural changes caused by the PS1 deletion, we analyzed a site-specific mutation at the tip of the PS1 hairpin that was predicted to sterically

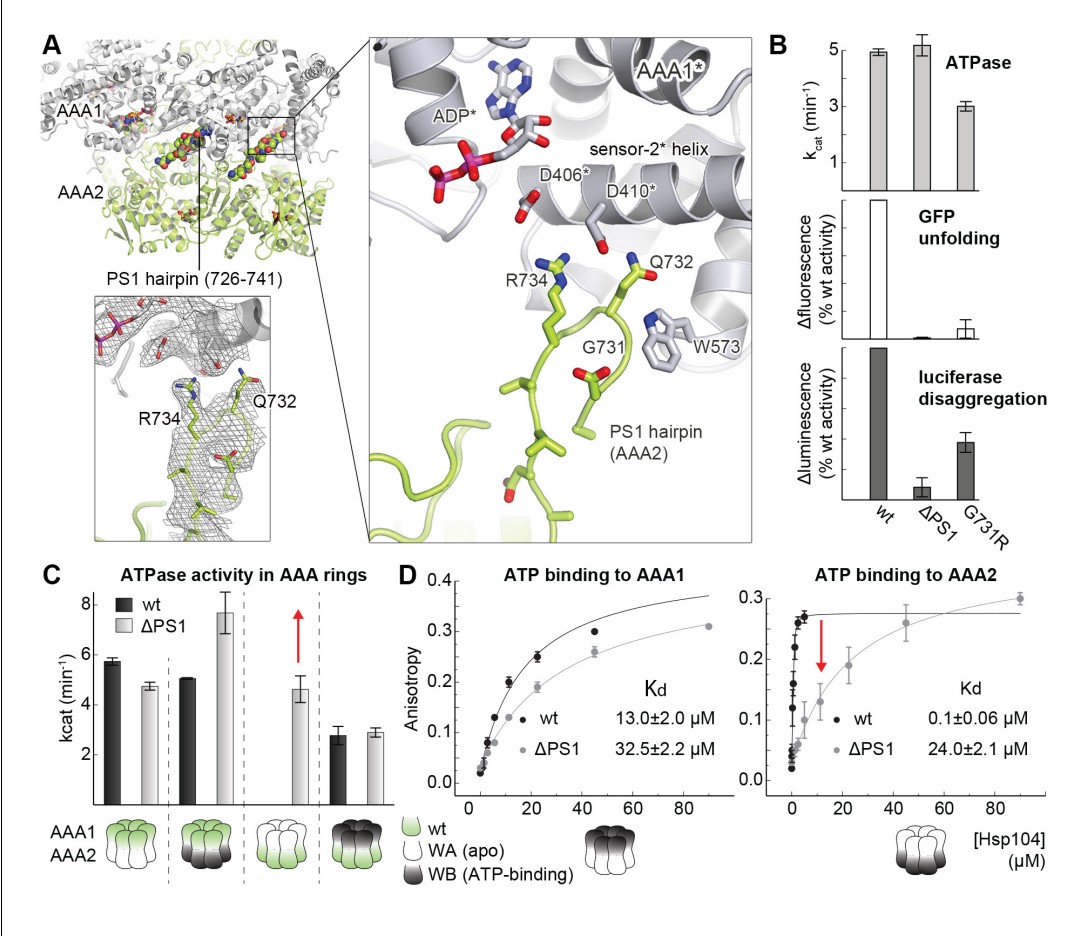

**Figure 6.** Functional coupling of the two AAA rings of Hsp104. (**A**) The PS1-hairpin of AAA2, which was well-defined by electron density (inset: omit density map contoured at 1.0 σ), forms specific contacts within the AAA1* active site. Bound ADP and interacting residues are shown in stick mode. (**B**) Characterization of the PS1 deletion (ΔPS1) and the G731R mutant showing that the PS1-hairpin is essential for unfoldase and disaggregase activity, but not for ATPase activity. (**C** and **D**) ATPase and mant-ATP binding assays reveal the role of the PS1-hairpin in adjusting the activities and nucleotide binding affinities of AAA1 and AAA2 to each other. Strongest effects of the ΔPS1 mutation are highlighted (red arrow). The used AAA variants (WA/WB combined with wildtype) are indicated. Error bars represent standard deviations.

The following figure supplements are available for figure 6:

**Figure supplement 1.** Position of the PS1-hairpin in hexameric Hsp104.

**Figure supplement 2.** Effect of PS1-hairpin deletion on ScHsp104 activity.

**Figure supplement 3.** SEC profiles of CtHsp104 dWA and dWB mutants.

**Figure supplement 4.** Effect of casein on ATPase activity.

expel the PS1 motif from the active site of AAA1. For this purpose, we replaced Gly731, which is in close contact to Trp573, by arginine. When tested in our activity assays, the G731R mutant had a slightly decreased ATPase activity, but it was even more impaired in its unfoldase and disaggregase activity, thus mimicking the PS1 deletion phenotype (*Figure 6B*). The in vivo relevance of the described PS1 interactions is also emphasized by a reported mutation in the arginine residue at the tip of the PS1-hairpin in ClpB from *Arabidopsis thaliana* that, despite its conservative character (R705K), led to a loss-of-function phenotype (*Lee et al., 2005*).

To explore how the PS1-hairpin influences the communication between the two AAA rings, we analyzed the ATPase activities of the AAA1 and AAA2 engines separately. For this purpose, we combined Walker A (WA) or Walker B (WB) mutations (*Figure 6—figure supplement 3*) – which mimic the *apo* and ATP-bound states, respectively – in one AAA domain with a wild-type active site in the partner AAA domain. Using this approach, we observed a strong coupling between the AAA domains, especially the dependence of AAA2 activity on the nucleotide state of AAA1. Under the conditions where ATP is stably bound to AAA1 (WB mutation), the AAA2 ring remains fully active. In contrast, preventing ATP binding to the AAA1 ring (WA mutation) turns AAA2 activity off. Strikingly, deleting the PS1-hairpin abolishes this correlation and renders the AAA2 ring active independent of the nucleotide-binding state of AAA1. Of note, the same effect was observed in the presence of substrate proteins, emphasizing the role of the PS1 hairpin in coupling the two AAA engines during substrate translocation (*Figure 6C* and *Figure 6—figure supplement 4*). Moreover, the PS1-hairpin is critical for adjusting the nucleotide-binding properties of AAA1 and AAA2, which is particularly evident from the analysis of ATP binding to AAA2 (*Figure 6D*). Using the AAA1 WA mutant, the observed affinity of AAA2 for ATP was very high (Kd ≈ 0.1 µM), most likely due to an allosteric effect of an empty AAA1 on AAA2, as previously proposed (*Franzmann et al., 2011*). Removal of the PS1-hairpin decreases the affinity under these conditions by more than two orders of magnitude, suggesting that the allosteric control of AAA2 by AAA1 has now been compromised. Taken together, our structural and biochemical data demonstrate that the PS1 motif is the structural element coordinating the two ATPase engines of Hsp104.

## Signaling path between AAA1 and AAA2

Notably, the PS1-hairpin and the AAA2 nucleotide binding site are located on opposite sides of the central β-sheet. We therefore analyzed how the signal could be passed across the β-sheet. Two features evident in the Hsp104 crystal structure seem to be important for this signaling. First, the PS1-hairpin is directly connected via strand β4 with the sensor-1 motif of AAA2 (Asn748) that is critical for ATP hydrolysis (*Hattendorf and Lindquist, 2002b*). Secondly, the central β-sheet of AAA2 appears to be slightly distorted in the Hsp104 filament. Structural comparison with the isolated AAA2 domain from ClpB (*Zeymer et al., 2014*) suggests that in the Hsp104 oligomer, contacts to the AAA1 domain, as well as to neighboring AAA2 subunits, induce a distortion of the central β-sheet, in particular of the two capping strands (*Figure 7A* and *Figure 7—figure supplement 1*). Notably, parallel β-sheets are presumed to be less stable than the anti-parallel counterparts (*Richardson, 1977*) and may thus be amenable to such deformation. Furthermore, the entrance point of the PS1-hairpin into the β-sheet of AAA2 is lined on one side by the strictly conserved Pro627, which does not form hydrogen bonds with the backbone of strand β4. We hypothesize that, owing to the observed imperfect secondary structure, the middle strand β4 may undergo small rearrangements within the frame of the central β-sheet. Such mobility would allow the repositioning of the sensor-1 residue Asn748 (AAA2) via β4/PS1 in response to subunit reorientation in the AAA1 ring (*Figure 7B*). To ensure efficient coupling, the β4/PS1 mechanical link should have a defined conformation. To test this prediction, we inserted a single Gly residue between residues Asn740 and Cys741 that directly precede strand β4. In addition to this PS1+ decoupling mutant, we shortened the PS1-hairpin by two residues (Δ737-738, PS1- mutant) or modified the central β-sheet of AAA2 by mutating the Pro-Pro-Ser motif of strand β1 by a less distorting sequence (P627T-P628G-S629N, PP mutant). Similar to the ΔPS1 deletion, these mutations led to a severe reduction in the unfolding and disaggregation activities, while the ATPase activity of AAA1 and AAA2 became partially uncoupled (*Figure 7B* and *Figure 7—figure supplement 2*). These data underscore the tight spatial constraints underlying the PS1-mediated coupling of the two ATPase rings. To test constraints imposed at the opposite side of the signaling path, i.e. at the sensor-1 residue of AAA2 located at the distal end of β4, we mutated Asn748 to either a shorter (Ser) or a longer (Gln) amino acid with similar hydrogen bonding properties. Although mutating the sensor-1 residue of AAA2 had only a minor influence on the overall ATPase activity (*Figure 7C*), the two mutations clearly affected the nucleotide binding in the remote AAA1 ring. When AAA2 was present in the apo state (WA), both sensor-1 mutations markedly increased the binding affinity of ATP to AAA1 (*Figure 7C*). This experiment demonstrates that changes in the sensor-1 residue can be communicated to AAA1 and support our hypothesis that the sensor-1 residue of AAA2 is part of the PS1 signaling device synchronizing the activities of the two AAA rings. It should be also noted that the N748S and N748Q mutations had opposite

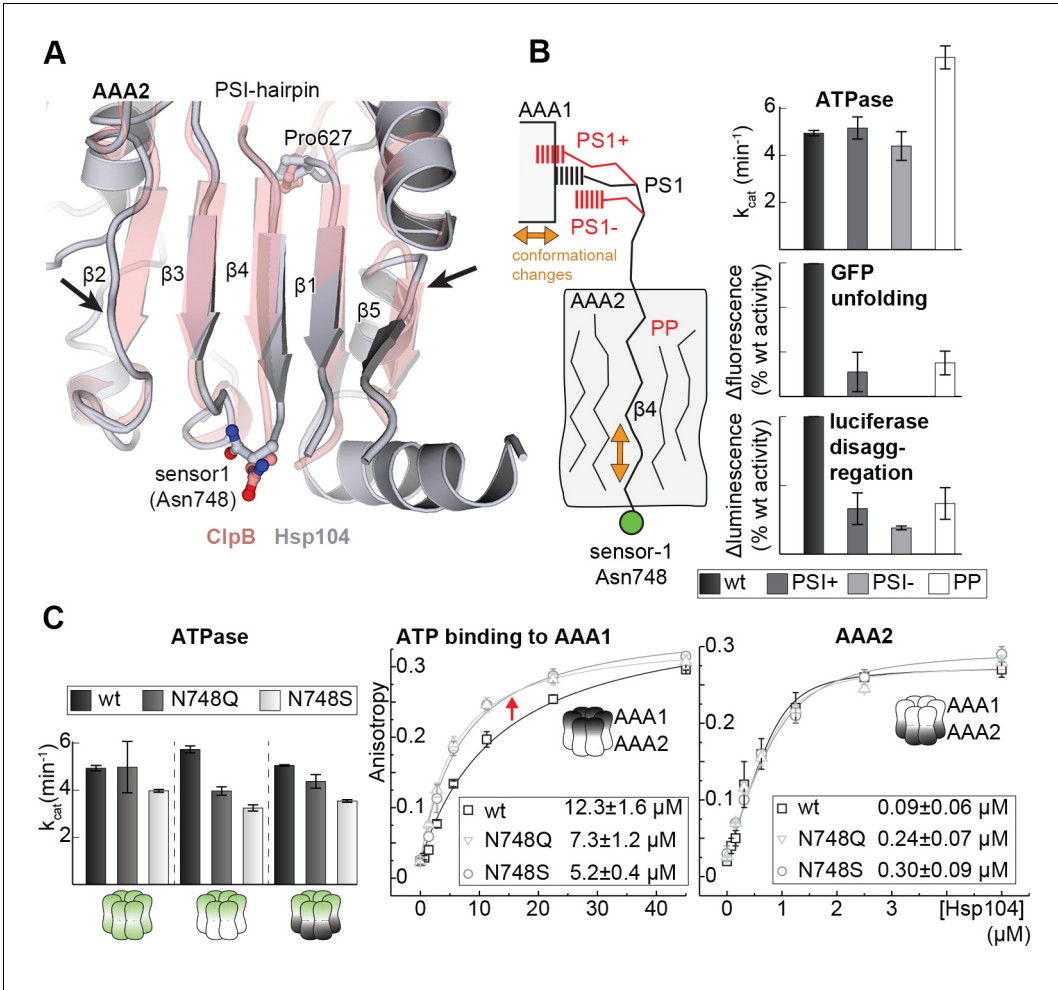

**Figure 7.** Signal path between AAA1 and AAA2. (**A**) Superposition of the AAA2 domain of the Hsp104 filament (grey) with the AAA2 domain of ClpB that was crystallized as isolated domain (salmon; PDB 4lj5). The ribbon model illustrates the distortion of strands β2 and β5 (arrows) and the shorter strands β1, β3, and β4 observed in the Hsp104 structure. See also *Figure 7—figure supplement 1*. (**B**) Model of how conformational changes of an AAA1 ATPase module or reorientations of strand β4 and the sensor-1 residue are communicated by the PSI-hairpin. Mutants predicted to decouple signaling (PS1+, PS1- and PP) were analyzed for their ATPase, unfolding, and disaggregation activities. See also *Figure 7—figure supplements 2* and *3*. (**C**) ATPase activity assay and mant-ATP binding data of sensor-1 mutants (N748S and N748Q). Strongest effects compared to the respective wildtype control are highlighted (red arrow). The used AAA variants (WA/WB combined with wildtype) are indicated. Error bars represent standard deviations.

The following figure supplements are available for figure 7:

**Figure supplement 1.** Signal path between AAA1 and AAA2.

**Figure supplement 2.** PS1+ and PP mutations partially uncouple the ATPase activity of AAA1 and AAA2.

**Figure supplement 3.** Activities of sensor-1 mutations.

effects on the overall unfoldase activity of Hsp104 (*Figure 7—figure supplement 3*). This puzzling result cannot be explained by an altered communication between AAA1 and AAA2, as both mutants caused similar enzymatic effects in AAA1 and AAA2, respectively (*Figure 7—figure supplement 3*). Owing to the close distance between the PS1 hairpin and the AAA2 pore-loop (*Biter et al., 2012b*), we presume that the opposite effects of the sensor-1 mutations on unfolding activity may reflect

different substrate translocation properties of the Hsp104 particles; however, the molecular mechanism of this intriguing function remains to be elucidated.

## Discussion

AAA molecular machines are only functional as oligomers and require coordination between the individual building blocks for efficient activity. While the homotypic interactions between the six ATPase subunits within a single AAA hexamer are increasingly well understood, the mechanistic importance of heterotypic communication is less clear and its molecular underpinnings were not resolved in the recent cryoEM analysis (*Yokom et al., 2016*). Heterotypic contacts include, for example, interactions with the ATPase subunits of a second AAA ring, with additional regulatory domains or proteins, and with cognate substrate molecules. The present study reveals two novel mechanisms how such *trans*-regulation is mediated in the AAA disaggregase Hsp104, mechanisms that may be generally applicable to AAA proteins of various functions and also, to some extent, to other multi-protein machineries.

### Steric control of complex AAA machines

Hsp104 and its orthologs are under control of their MD extension that can form a continuous ring around the hexamer. Binding of Hsp70 to the MD opens this ring and stimulates disaggregase activity by an unknown mechanism. Our data demonstrate that the MDs compose a topological belt tightly embracing the AAA1 ring (*Figure 8A*). Each MD glues two ATPase L/S* modules together, thus restricting their relative movement and keeping the AAA machine in a latent state. In this self-entrapment process, motif-2 serves as a molecular tether that physically links neighboring ATPase units via two well-defined binding sites. The flexible motif-1 associates with the solvent-directed face of motif-2 thereby closing the MD belt and stabilizing the MD/AAA1 interactions. As shown previously, the Hsp70 chaperone, functioning as a substrate-recruiting adaptor, competes with motif-1 for binding to motif-2. According to the modeled Hsp70/Hsp104 complex (*Rosenzweig et al., 2013*), Hsp70 binding would lead to a rearrangement of motif-2 disrupting its contacts with AAA1. The now released ATPase modules are free to fulfill their dynamic task in the protein unfolding and disaggregation reaction. As it is assumed that chaperones, unfoldases, and proteases acting on damaged proteins must be highly flexible to be active (*Saibil, 2013*), it will be interesting to see whether other quality control factors are regulated by similar restraint mechanisms or factors. To our knowledge, the motor protein dynein is the only other AAA enzyme employing a comparable steric brake. Dynein, which is composed of six fused ATPase domains, is regulated by a "doorstop" mechanism, whereby the regulatory protein Lis1 binds to the AAA ring and sterically hinders progression through the AAA mechano-chemical cycle (*Toropova et al., 2014*). Notably, restricting the conformational freedom of individual subunits seems to be an emerging theme in regulating large macromolecular machineries, seen for example also in the ribosome. Here, a chaperone complex implicated in co-translational folding of nascent polypeptides, RAC, physically links adjacent ribosomal subunits thus controlling the inter-subunit rotation required for peptide elongation (*Zhang et al., 2014*).

### Synchronizing ATPase engines in complex molecular machines

The Hsp104 crystal structure identifies the structural motif that mechanically links the two ATPase engines of a HSP100 chaperone machine. We suggest that the PS1-hairpin and the associated strand β4 couple conformational changes in AAA1 with the repositioning of the catalytic sensor-1 residue of AAA2 (*Figure 8B*). The signaling mechanism relies on the rearrangement of individual β-strands within the AAA2 β-sheet, which presumably requires a distortion of secondary structure as observed in the present Hsp104 crystal structure. We hypothesize that the relatively short parallel β-sheet found in the AAA fold is well suited for such a distortion. This may be one of the factors underpinning the dynamic and highly allosteric nature of AAA proteins. Aside from the inter-ring communication, the PS1 loop has been implicated in intra-ring signaling, due to its location near the pore-loop of AAA2 (*Biter et al., 2012b*). Accordingly, the PS1 motif might coordinate inter-ring structural changes with the movements of the AAA1 and AAA2 pore-loops.

Identification of the inter-ring coupling device also provides insight into the conservation of HSP100 remodeling proteins. A detailed sequence analysis showed that the PS1-hairpin, the

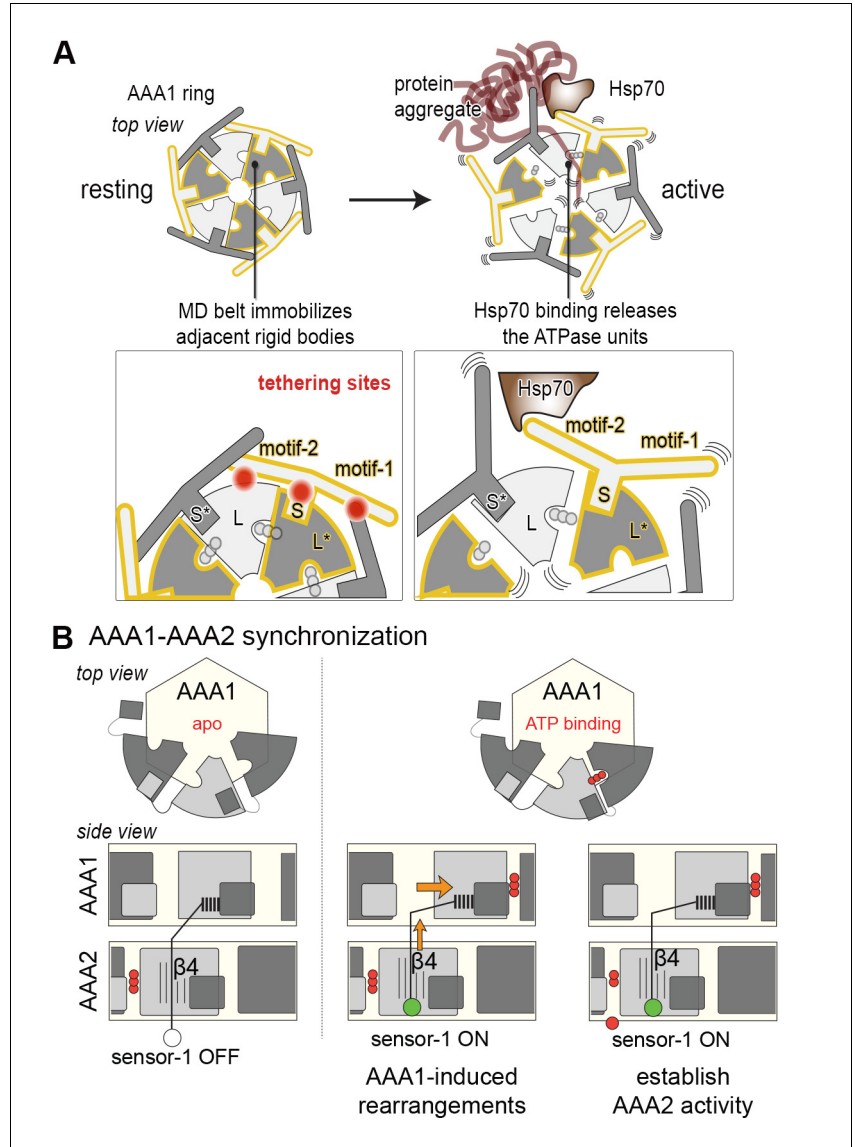

**Figure 8.** Novel mechanistic features of the Hsp104 disaggregase. (**A**) A restraint mask composed of the MD keeps the disaggregase inactive by immobilizing ATPase modules of the AAA1 ring. Binding of Hsp70 to the MD opens the safety belt and activates Hsp104 to act on the presented protein aggregate. The zoomed-in windows illustrate the multiple contact sites established by the MD that result in a physical tethering of adjacent rigid bodies (neighboring subunits are colored in different grey tones and rigid bodies are framed). (**B**) Allosteric coupling of the two AAA rings by the PS1 motif of AAA2. Nucleotide-dependent movements of an AAA1 module are transduced via the PS1-hairpin over a large distance leading to the repositioning of the catalytic sensor-1 residue of AAA2. As indicated, the functional switch relies on the reorientation of strand β4 in the AAA2 domain.

acceptor site at AAA1, and elements destabilizing the AAA2 β-sheet are among the most conserved sequence stretches in the HSP100 unfoldase/disaggregase family comprising for example ClpA, ClpB, ClpC, ClpE, and Hsp104 (*Supplementary file 2*). In contrast, Cdc48 and related enzymes, which are also composed of stacked AAA1 and AAA2 hexamers, lack these motifs, pointing to distinct inter-ring signaling mechanisms and functions. Finally, it should be noted that the PS1-hairpin is the defining feature of a major AAA superclade comprising also single-ring ATPase machines of various remodeling functions (*Erzberger and Berger, 2006*). Among these, a signaling link between PS1 and sensor-1 has been proposed to occur in DNA-remodeling proteins (*Schumacher et al., 2007*). We thus suggest that the PS1-hairpin, the associated β4-strand, and sensor-1 residue

constitute a common signaling device to couple ATPase activity with external stimuli such as substrate, co-factor, or ligand binding. It is this multi-purpose structural device sensing heterotypic contacts that provides the long-sought link between the two ATPase engines in the powerful protein unfolding machine Hsp104.

## Materials and methods

### Cloning, expression and purification

The *Saccharomyces cerevisiae* Hsp104 and *Escherichia coli* GroEL genes were cloned from yeast and bacterial genomic DNA, respectively. The genes of full-length Hsp104, Hsp70, and Hsp40 from *Chaetomium thermophilum*, as well as for the RepA-GFP fusion protein (70 initial amino acids of RepA followed by full-length GFP), were synthesized chemically. All constructs were cloned into the pET21a vector and expressed in *E. coli* yielding recombinant proteins with a C-terminal His6-tag. To generate Cys-Cys mutants for cross-linking experiments, we used a Cys-free, but fully functional Hsp104 variant, in which all Cys residues were replaced by Ser. All mutations were introduced by site-directed mutagenesis using specific DNA primers.

Except Hsp40, which was purified using a Zn-Sepharose column, all proteins were affinity-purified by NiNTA chromatography applying a step-wise imidazole gradient. In the second chromatography step, the samples were loaded onto a Resource Q anion exchange column and separated by a linear NaCl gradient. To transfer the purified proteins into buffer-A (20 mM HEPES, pH 8.1, 100 mM NaCl) supplemented with 1 mM TCEP, we used pre-equilibrated buffer-exchange and gel-filtration columns (Superdex-200 for Hsp104, GroEL, and Hsp70; Superdex-75 for Hsp40 and RepA-GFP). After concentrating proteins to about 0.1 mM, the samples were flash-frozen in liquid nitrogen and stored at −80°C until use.

### Crystallization and structure determination

Prior to crystallization trials, the full-length Hsp104 protein of *C. thermophilum* was incubated with an equimolar mixture of ADP and AlF$_3$ (3 mM each). Crystals were grown at room temperature by the sitting-drop vapor-diffusion method. Upon mixing 200 nL of Hsp104 (7.5 mg/mL) with 100 nL of a reservoir solution containing 8.3% penta-erythritol propoxylate and 0.1 M MES-NaOH pH 5.8, hexagonal crystals appeared after three days. The crystals belonged to the monoclinic space group P2$_1$ and contained three Hsp104 subunits per asymmetric unit. For cryo-protection, crystals were transferred to a solution containing 28% pentaerythritol propoxylate and 0.1 M MES-NaOH, pH 5.8, and subsequently flash-frozen in liquid nitrogen. Diffraction data were collected at beamline P14 at DESY (Deutsches Elektronen-Synchrotron, Hamburg, Germany). Data were processed with XDS (*Kabsch, 2010*) and scaled with SCALA (*Winn et al., 2011*). The structure was solved by molecular replacement using the program Phaser (*Mccoy et al., 2007*). As search models we used the AAA1 (PDB 4hse [*Zeymer et al., 2013*]) and AAA2 (PDB 1qvr [*Lee et al., 2003*]) domains of ClpB that were adapted to the Hsp104 sequence by MODELER (*Eswar et al., 2006*) and CHAINSAW (*Schwarzenbacher et al., 2004*). Model building and refinement proceeded in repeated cycles using the programs O (*Jones et al., 1991*), CNS (*Brünger et al., 1998*), and PHENIX (*Adams et al., 2010*). The final structure was refined at 3.7 Å resolution to an R-factor of 23.7% (R$_{free}$ value of 27.7%) with Ramachandran statistics having 86.8% of the residues in the favored region and 0.2% in the disallowed region. Data collection, phasing, and refinement statistics are summarized in *Table 1*.

### Sequence alignments

Sequence alignments of Hsp100 disaggregases were performed with the program MAFFT (version 6, L-INS-I method) (*Katoh and Toh, 2008*) and visualized using the ESPript 3.0 server(*Robert and Gouet, 2014*). Sequences are derived from the NCBI protein database: Hsp104_Ct: *Chaetomium thermophilum* (gi|340959261|); HS104_YEAST: *Saccharomyces cerevisiae* (gi|6323002|ref| NP_013074.1|); Hsp104_Um: *Ustilago maydis* (gi|71024695|); CLPB_ECOLI: *Escherichia coli* (gi| 15832709|); CLPB_THET8: *Thermus thermophilus* (gi|55981456|); HSP78_YEAST: *Saccharomyces cerevisiae* (gi|398366295|); CLPC_BACSU: *Bacillus subtilis* (gi|16077154|), CLPE_BACSU: *Bacillus subtilis* (gi|16078434|ref|NP_389253.1|); CLPA_ECO57: *Escherichia coli* (gi|15830222|). For coloring the structure according to conservation scores, Hsp104 sequence orthologs were first retrieved from the

NCBI non-redundant protein database using NCBI-BLAST (version 2.2.26, *E*-values < 1e-180) (*Altschul et al., 1997*) and then 20 fungal sequences representing a wide taxonomic range were selected. Sequence conservation values were calculated with the program al2co, using no weighting scheme for amino acid frequency estimation, the sum-of-pairs measure conservation calculation method, and the BLOSUM62 scoring matrix (*Pei and Grishin, 2001*).

## Modeling of the Hsp104 hexamer

To model the planar Hsp104 hexamer we used a single L/S* building block composed of the AAA1L/AAA1S* and AAA2L/AAA2S* domains of two adjacent protomers of the crystallized filament. The hexameric Hsp104 was generated by overlying the AAA1 part of the L/S* module onto the six AAA1 modules of the hexameric ClpC structure (PDB 3pxi (*Wang et al., 2011*)). Structural alignments were done with O (*Jones et al., 1991*) and the model was energy-minimized by CNS (*Brünger et al., 1998*). To generate the overlay with the ScHsp104 EM structure (*Yokom et al., 2016*) the same strategy was applied and the AAA1L/AAA1S* rigid bodies were aligned using the program Pymol (*Delano, 2002*).

## Cross-linking coupled mass spectrometry (XL-MS)

Cross-linking was performed by mixing 0.7 mg/mL Hsp104 with various amounts of an equimolar mixture of isotopically light (d0) and heavy (d12) labeled BS$^3$. The reaction mixture was incubated for one hour at room temperature and subsequently quenched by the addition of 100 mM ammonium bicarbonate. The appropriate BS$^3$ concentration (repressed variant and wild-type 0.3 mM, hyperactive variant 0.6 mM, respectively) was determined based on SDS-PAGE (*Figure 4—figure supplement 1*) and the chemical cross-links on Hsp104 were identified by mass-spectrometry as previously described (*Herzog et al., 2012*). Briefly, proteins were denatured by the addition of two sample volumes of 8 M urea and reduced with 5 mM TCEP for 20 min at 35°C. Subsequently, proteins were alkylated with 10 mM iodoacetamide and incubated for 40 min at room temperature in the dark. Digestion of the cross-linked proteins was performed with lysyl endopeptidase at an enzyme ratio of 1 to 50 (w/w) at 37°C for 2 hr. A second digestion with trypsin (also at 1:50 ration w/w) was completed at 37°C overnight. Cross-linked peptides were enriched by size exclusion chromatography on a Superdex Peptide PC 3.2/30 column (300×3.2 mm). The cross-link fractions were analyzed by liquid chromatography coupled to tandem mass-spectrometry using a LTQ Orbitrap Elite instrument. Cross-linked peptides were identified using the *xQuest* (*Walzthoeni et al., 2012*) software and cross-links were visualized by the *xVis* server (*Grimm et al., 2015*). False discovery rates (FDRs) were estimated by the program, *xProphet* (*Walzthoeni et al., 2012*) and results were filtered according to the following parameters: FDR = 0.05, min delta score = 0.90, MS1 tolerance window of −4 to 4 ppm, ld-score > 22.

## Cys-Cys cross-linking

To ensure efficient Cys-Cys cross-linking, proteins were oxidized by the addition of 25 μM dichloro (1, 10-phenanthroline) copper. After 15 min, the reactions were stopped by transferring the reaction mixture to fresh buffer-A. To insert the maleimide or the bismaleimidoethane (BMOE) cross-linker, the Cys-Cys variants of Hsp104 were first treated with 10 mM DTT to reduce potential disulfides. After 5 min, proteins were transferred into buffer-B (20 mM HEPES, pH 7.5, 100 mM NaCl, 5 mM EDTA) using a PD10 desalting column. Reactions were started by adding a two-fold molar excess of BMOE (or maleimide) and stopped after 15 min by the addition of 10 mM DTT. Upon a buffer exchange to buffer-A containing 1 mM TCEP, the cross-linking efficiencies were characterized by non-reducing SDS-PAGE analysis.

## ATPase assay

ATPase activity was determined by a coupled enzymatic reaction (*Nørby, 1988*). 1.5 μM Hsp104 were incubated with 37.5 U/mL pyruvate kinase, 42.9 U/mL lactacte dehydrogenase, 0.25 mM NADH, 15 mM phosphoenolpyruvate, 5 mM MgCl$_2$ and varying ATP concentrations. $A_{340}$ was recorded for 10 min using a Synergy H1 Multi-Mode Reader. Experiments were repeated three times using protein from independent purifications. The molar ATPase activity ($k_{cat}$) was calculated by the following equation:

$$\text{kcat} = \frac{\Delta A_{340}}{path\ length \times 6220\ \text{M}^{-1} \times \text{cm}^{-1} \times [\text{Hsp104}]}$$

Resulting activities were plotted against the ATP concentration and kinetic parameters were fitted using the Hill equation in the program ORIGIN (*Seifert, 2014*).

### RepA-GFP unfolding assay

The unfolding activity of Hsp104 was assayed by monitoring the decrease in fluorescence of the model substrate RepA-GFP (*Doyle et al., 2007b*). To this end, 10 µM Hsp104 were pre-incubated with 2.1 µM GroEL-trap (GroEL D87K), 300 nM RepA-GFP, the ATP regeneration system (15 U/mL pyruvate kinase, 6.25 mM phosphoenolpyruvate), and 0.1 mg/mL BSA in buffer-A supplemented with 1 mM TCEP, 0.1 mM EDTA, 10% (v/v) glycerol, and 10 mM MgCl$_2$. The reaction was started by adding an equimolar mixture of ATP and ATPγS (5 mM each). The change in fluorescence was monitored at $\lambda_{ex}$ = 395 nm and $\lambda_{em}$ = 509 nm, at room temperature. Unfolding activities were derived from the decrease of the fluorescence signal within the initial 2–10 min time window. All experiments were performed as triplicates using protein from independent purifications. Although experiments performed in the presence of ATP/ATPγS reflect the unfolding mechanism mediated by Hsp104 only partially (*Kummer et al., 2016*), they do represent a valuable tool to compare the activity of various Hsp104 mutants independent of their ability to cooperate with Hsp70.

### Luciferase disaggregation assay

As a model substrate to measure protein disaggregation, we used firefly luciferase. To generate the aggregated substrate, luciferase (0.5 µM) was dissolved in buffer-C (25 mM HEPES pH 7.5, 150 mM KCl, 15 mM MgCl$_2$) and heat-denatured at 45°C. For the disaggregation reaction, 1.5 µM Hsp104 was mixed with 1.5 µM Hsp70, 1.5 µM Hsp40, 0.05 µM aggregated luciferase, 5 mM ATP, the ATP regeneration system (15 U/mL pyruvate kinase, 6.25 mM phosphoenolpyruvate), and 0.1 mg/mL BSA in buffer-C. After 90 min of incubation at 30°C, luciferin was added to the final concentration of 15 µM and luminescence was recorded with a PHERAStar plate reader. Reactions lacking Hsp104 were used as a negative control. The average refolding efficiency for wt CtHsp104 was between 5% and 15%. All experiments were performed three times using protein from independent purifications.

### Fluorescence anisotropy measurements

Experiments were performed in triplicate using inactive protein (WA and WB mutants) from independent purifications. For all titrations, mantATP was kept constant at 1 µM. Fluorescence anisotropy measurements were carried out at room temperature using $\lambda_{ex}$ = 355 nm and $\lambda_{em}$ = 448 nm. Observed anisotropy values (*A*) were plotted as a function of Hsp104 concentration and fitted to the following equation with ORIGIN:

$$A = A_0 + (A_1 - A_0) \times \left( ([\text{mantATP}] + \text{K}_d + [\text{Hsp104}]) - \sqrt{\frac{(-[\text{mantATP}] - \text{K}_d - [\text{Hsp104}])^2 - 4 \times [\text{mantATP}] \times [\text{Hsp104}]}{2[\text{mantATP}]}} \right)$$

$A_0$: anisotropy observed with free mantATP; $A_1$: anisotropy of protein bound mantATP
$K_d$: equilibrium dissociation constant

## Acknowledgements

We thank Janine Kirstein for advice in assay design and Gleb Bourenkov and Thomas Schneider at DESY for assistance during data collection. This work was supported by the Austrian Research Promotion Agency (FFG). The IMP is funded by Boehringer Ingelheim.

# Additional information

## Funding

| Funder | Grant reference number | Author |
|---|---|---|
| Boehringer Ingelheim | IMP is funded by Boehringer Ingelheim | Alexander Heuck<br>Sonja Schitter-Sollner<br>Marcin Józef Suskiewicz<br>Robert Kurzbauer<br>Juliane Kley<br>Alexander Schleiffer<br>Tim Clausen |
| European Molecular Biology Organization | Long-Term Fellowship ALTF 1078-2010 | Alexander Heuck |
| Austrian Science Fund | Hertha-Firnberg fellowship T557-B11 | Sonja Schitter-Sollner |
| European Research Council | StG no. 638218 | Franz Herzog |
| Österreichische Forschungs-förderungsgesellschaft | No 852936 | Tim Clausen |
| European Research Council | AdG no. 694978 | Tim Clausen |

The funders had no role in study design, data collection and interpretation, or the decision to submit the work for publication.

## Author contributions

AH, SS-S, MJS, Conception and design, Acquisition of data, Analysis and interpretation of data, Drafting or revising the article; RK, JK, Conception and design, Acquisition of data, Analysis and interpretation of data; AS, FH, Acquisition of data, Analysis and interpretation of data, Contributed unpublished essential data or reagents; PR, Conception and design, Acquisition of data, Contributed unpublished essential data or reagents; TC, Conception and design, Analysis and interpretation of data, Drafting or revising the article

## Author ORCIDs

Marcin Józef Suskiewicz, http://orcid.org/0000-0002-3279-6571
Tim Clausen, http://orcid.org/0000-0003-1582-6924

# Additional files

## Supplementary files

• Supplementary file 1. Published Hsp104 and ClpB mutants with repressive or hyperactive phenotype.

• Supplementary file 2. Sequence alignment of Hsp104 and related HSP100 disaggregases.

## Major datasets

The following dataset was generated:

| Author(s) | Year | Dataset title | Dataset URL | Database, license, and accessibility information |
|---|---|---|---|---|
| Heuck A, Schitter-Sollner S, Clausen T | 2015 | Crystal Structure of Hsp104 form C. thermophilum | http://www.rcsb.org/pdb/search/structid-Search.do?structureId=5d4w | Publicly available at RSCB PDB (Accession no: 5d4w) |

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
