## [Decision Letter]

[Editors’ note: a previous version of this study was rejected after peer review, but the authors submitted for reconsideration. The first decision letter after peer review is shown below.]

Thank you for submitting your work entitled "Structural basis for the disaggregase activity and regulation of Hsp104" for consideration by *eLife*. Your article has been evaluated by Michael Marletta (Senior Editor) and four reviewers, one of whom is a member of our Board of Reviewing Editors. The reviewers have opted to remain anonymous.

Our decision has been reached after consultation between the reviewers. Based on these discussions and the individual reviews below, we regret to inform you that your present submission will not be considered for publication in *eLife*. However, you will see that the reviewers, although raising substantial technical concerns, found your work interesting and important for the field. We therefore encourage you to consider resubmission once you have been able to address the referees' comments.

The referees expressed concern regarding the quality of the structural model (reviewer #1), interpretation of the crosslink data (reviewers #2 & #4), controls for various biochemical experiments (reviewer #3) and the correct citation of the literature (reviewer #2).

*Reviewer #1:*

Hsp104 is the fungal homolog of the bacterial ClpB and ClpC proteins. These hexameric molecular machines consist of two AAA (AAA1 and AAA2) rings. A characteristic feature is a coiled-coil insertion (CCD) in the AAA1 ring. The α-helical propeller-like structure of the CCD is important for functional cooperation with Hsp70 in protein disaggregation. In fungi, Hsp104 is a crucial component of the response to proteotoxic heat stress. Extensive structural and mechanistic studies of this class of molecular chaperones have previously been published. Crystal and cryo-EM structures exist for ClpB/ClpC and Hsp104, respectively. Numerous mutants, especially relating to the function of the CCD, have been characterized. However, questions about the regulation and the mechanism remained unanswered.

The present manuscript reports the first crystal structure of the eukaryotic disaggregase protein Hsp104. The crystal lattice does not contain the functionally active hexamer, but rather a helical filament of subunits. However, the structural building block, a complex between large and small AAA subdomains, resembles closely the respective units in the hexamer structure of ClpC. I am surprised about the poor stereochemical quality of the model, with 82.6, 13.3 and 4.1% of the peptide bonds in favored, allowed and disallowed regions of the Ramachandran plots, respectively. For an excellent-quality model >98, <2 and <0.05% are expected. This suggests serious over-fitting of the data, thereby distorting the stereochemistry, and thus casts doubt on the atomic presentation/discussion of the biochemically important contacts. Restraining the stereochemistry more tightly and accepting slightly worse R-factors should yield a more plausible model. I also wonder whether the nominal resolution was properly chosen. The authors should consider using available automated tools like Rosetta_refine to improve the refinement of their model.

The biochemical evidence presented for a CCD belt being a "straitjacket" for the AAA1 domain mobility appears solid. However, it is strange that the reduced Cys-Cys mutant of Hsp104 (Figure 5) should yield much lower activity than wildtype and the BMOE-linked variant. The proposed model appears to be in line with previously published biochemical data.

In the Hsp104 crystal structure, the PS1-hairpins of the AAA2 ring reach out to nucleotide binding pockets in the AAA1 ring. This led the authors to suggest that the PS1-hairpins in the AAA2 domain regulate the allosteric communication between the AAA rings. Deletion of the PS1-hairpin or modification of the length between PS1 tip and sensor-1 motif of AAA2 seem to decouple ATPase from remodeling activity. These findings strongly support their hypothesis. Overall the manuscript is well-written and defines interesting mechanistic aspects of the Hsp104 machinery.

*Reviewer #2:*

Hsp104 is an AAA+ ATPase, which together with Hsp70/40 chaperones recovers functional protein from aggregates (Glover and Lindquist, 1998). Hsp104 orthologs are found in bacteria (e.g. TtHsp104 and EcHsp104), yeast (ScHsp104), and plants (e.g. AtHsp104), which are essential to the respective heat-shock responses, and share the ability to disaggregate amorphous protein aggregates. In addition to its role in the heat-shock response, ScHsp104 is also required for the propagation and maintenance of all amyloid-forming prions in *S. cerevisiae* (Chernoff et al., 1995), an activity that was presumed to be unique to ScHsp104 (Shorter and Lindquist, 2004). Hence, it was proposed that yeast and bacterial Hsp104 are different proteins and consequently must differ in their three-dimensional structure. This resulted in the infamous model by Wendler et al., who proposed that ScHsp104 is an atypical AAA ATPase, which differs in structure from bacterial Hsp104 and other AAA+ ATPases (Wendler et al., 2007; Wendler et al., 2009). It is now firmly established, but not uniformly accepted, that yeast and bacterial Hsp104 are conserved in both structure (Carroni et al., 2014; Lee et al., 2010; Sweeny et al., 2015) and function (Reidy et al., 2012; Yuan et al., 2014). It is worth noting that the highly speculative and incorrect Wendler structure is now refuted by the same groups (Carroni et al., 2014; Sweeny et al., 2015; Wendler et al., 2012), but continues to be cited in the current literature causing major damage to the field.

Heuck et al. reports the first crystal structure of a fungal Hsp104. The work is a welcome addition to the debate, and reconfirms that fungal Hsp104 (CtHsp104) is structurally conserved with bacterial Hsp104 (Carroni et al., 2014; Lee et al., 2003). Like the previously reported crystals structures of bacterial Hsp104, CtHsp104 crystallized as a helical filament, but is believed to form a hexameric ring-assembly similar to other single- and double-ring Hsp100 ATPases (Glynn et al., 2009; Sousa et al., 2000; Wang et al., 2011; Wang et al., 2001a; Wang et al., 2001b). Heuck et al. confirmed this using cross-linking coupled mass spectrometry and site-specific disulfide crosslinking. Finally, the authors proposed an elegant mechanism how the ATP signal is transduced between the AAA-1 and AAA-2 ring, involving a PS1 hairpin that may function as a signal transducer.

On balance, the data support the conclusion drawn. Although the work is a welcome addition and may be suitable for publication, is requires major revisions. Especially care should be taken when revising the manuscript because some of the references are incorrect or misleading, while others are missing. A re-review of the revised manuscript is advisable due to the collection of minor issues that distract from the main thrust of the current paper.

1) It is unfortunate that the crystal structure of CtHsp104 and not of ScHsp104 was determined, which does not directly address the controversy that has been damaging the field. For instance, the biochemical activities of CtHsp104, while similar to ScHsp104, are not identical, and could fuel the argument that CtHsp104 is different from ScHsp104. Is CtHsp104 essential for fungal prion propagation?

2) A major limitation of the present work is that the X-ray structure of the ring assembly was not determined. Moreover, neither the N-domain nor C-terminal extension were resolved in the present structure, perhaps as a result of finding a structure solution by molecular replacement, arguing for independent phase information.

3) The authors do not make it clear how many of the cross-linked Lys, if any, disagree with their proposed model of active and inactive states. It would seem that interpretation of the cross-linking pattern is complicated since the structure of the active state was not determined. If the CCD represents a "restraint mask" or negative regulator of Hsp104, elimination of the CCD should activate the protein disaggregase. Yet, Hsp104 variants that lack the CCD are inactive in protein disaggregation. How can this be reconciled?

4) Care needs to be taken when reporting WA and/or WB Hsp104 mutants. It is known that WA mutants cause deoligomerization, while WB mutants can bind ADP or contain no nucleotide. At a minimum, the authors need to show that the mutants form hexamers.

5) The novelty of the PS1-hairpin of AAA-2 is somewhat overstated since it is identical to the previously reported β-hairpin that is part of the intersubunit signaling motif (Biter et al., 2012).

Additional comments:

1. It is now generally accepted that the Wendler model (Wendler et al., 2007) is wrong, falling short of a retraction. It is therefore confusing what the authors consider "consistent" with the Wendler model (e.g. subsection “Crystal structure of the Hsp104 subunit”, last paragraph). Moreover, it is doubtful that the Wendler structure has the necessary resolution to assign side-chain conformations correctly. Great care needs to be taken when using the terms "flexible", "motions", "structural rearrangements, which are loosely defined and refer to very different things in high-resolution X-ray structures and in structural work based lower resolution cryoEM, SAXS, etc.

2) Introduction: "Mixed L/S* modules…represent the functional units of AAA chaperones[…]". The large and small subdomains are both required for nucleotide binding. Neighboring subdomains may exert regulatory function but are dispensable for nucleotide binding in cis. This statement is misleading and needs to be revised.

3) Abstract: What is exactly meant by "near-atomic"? Does it refer to resolution (i.e. 3.7A) or coordinates (i.e. atoms can be seen)? Near-atomic is not appropriate to describe either (c.f. Results, first paragraph).

4) Introduction, first paragraph: It is true that AAA ATPases are found in cells of all organisms, but Hsp104 disaggregases are not present in animal cells, despite functional conservation of a protein disaggregating activity in higher Eukarya. This needs to be clearly distinguished.

5) Introduction, second paragraph: The present work is concerned with the structure-function analysis of *C. thermophilum* Hsp104. CtHsp104 and ScHsp104 are similar but non-identical (e.g. Figure 1). Also, in Figure 1 (right panel), it is unclear whether the luminescence measurements represent absolute or relative activities. What is the activity of the cognate ScHsp104:Hsp70/40 system? What is the SEM?

6) Introduction, second paragraph: The ability of Hsp104 to unravel prions on its own has been questioned (Inoue et al., 2004; Krzewska and Melki, 2006; Reidy et al., 2012; Yuan et al., 2014). It is now widely accepted that the ability to dissolute prions is dependent on the Hsp104-Hsp70 bi-chaperone system, and is the same for the dissolution of for amorphous aggregates.

7) Introduction, second paragraph: Rosenzweig et al. did not demonstrate that Hsp70 binding to the M-domain activates the Hsp100 motor (Rosenzweig et al., 2013). They showed that DnaK binds to the M-domain of Hsp100 confirming earlier work by Haslberger et al. (Haslberger et al., 2007).

8) Introduction, end of second paragraph: It is now widely accepted that the Wendler structure (Wendler et al., 2007) is incorrect and contrasts the more recent work by the same authors (Carroni et al., 2014). This statement needs to be revised. In this context, recent work by the Bukau, Wickner and other labs how the CCD regulates the Hsp100 motor should be discussed.

9) Subsection “Crystal structure of the Hsp104 subunit”, last paragraph: How was the identity of sensor-2 defined? There should only be one, not three.

10) Subsection “Crystal structure of the Hsp104 subunit”, last paragraph: "[…]Arg349 has been previously described as the Arg finger[…]? What is the reference for this?

11) Subsection “ATPase rigid bodies are maintained in the crystallized Hsp104 filament”, last paragraph: Glynn et al. (Glynn et al., 2009) reported the hexameric crystal structure of ClpX, a single-ring Hsp100 member. This statement needs to be rephrased. Also, what about ring-forming structures determined by cryoEM?

12) [Supplementary-material SD3-data]: The sequence numbering does not match with the sequence numbering of the atomic coordinates.

*Reviewer #3:*

AAA proteins are mechanochemical ATPases that perform conformational work fueled by ATP hydrolysis to remodel substrates. The yeast AAA protein Hsp104 functions as a disaggregase by reactivating aggregated proteins in cooperation with Hsp70. Hsp104 consists of two AAA domains (AAA-1, AAA-2) and a regulatory coiled-coil domain (CCD, also termed M-domain). How ATPase activity is regulated and how the two ATPase modules communicate is of central importance for understanding disaggregase mechanism. Here the authors determined the crystal structure of *Chaetomium thermophilum* Hsp104 providing them a structural map to dissect the regulation of ATPase and disaggregation activity.

The determined crystal structure is similar to the ones of bacterial ClpB homologs. However, it includes additional novel and valuable information as crucial contacts between neighboring AAA domains are maintained in contrast to former structures. The novel Hsp104 structure confirms the previously established position and regulatory function of the CCD yet the interaction details are of higher resolution. The authors show that the CCD is restricting AAA-1 mobility, providing a rationale for down regulation of ATPase and disaggregase activity by the CCD. This part of the study is well performed and includes novel information, yet it also represents an evolution of an already established and accepted mode of Hsp104 activity control. In the second part, the authors identify a conserved structural element, the PS1 hairpin, which is suggested to act as sensor mediating interdomain communication between AAA-1 and AAA-2. A role of the hairpin in controlling ATPase and disaggregase activity is supported by biochemical analysis, though some assays and results need further clarification (see below). How exactly PS1 synchronizes the AAA modules remains vague and therefore some of the conclusions should be softened.

Overall the presented work is of high quality, contributes to an improved understanding of disaggregase mechanism and includes the identification of a novel regulatory element. The following points should be addressed in a revised manuscript:

1) The RepA-GFP unfolding assay requires the presence of non-physiological ATP/ATPγS mixtures. The relevance of such conditions for disaggregase function (disaggregation activity in presence of these nucleotide mixtures is low or not existing) and ATPase communication is unclear and results based on the assay are at least in parts questionable.

2) ATPase activities were always determined in the absence of substrate. Casein and specific peptides stimulate ATPase hydrolysis by Hsp104. It is recommended to include those in ATPase measurements as it might allow to more precisely defining the role of PS1 in signal transduction.

3) To exclude that deletion of the PS1 hairpin causes structural defects it is recommended to analyze the effect of e.g. Q732/R734 mutation, which should abolish interaction with the sensor 2 helix.

4) Figure 7: The two analyzed variants of N748 (sensor1) have opposing effects on GFP unfolding (Figure 4), a conflict overlooked and not discussed so far.

5) While the data demonstrate that PS1 plays a role in ATPase communication, the precise mechanism remains unclear. In their model (Figure 8) the authors suggest a defined order of signaling events between the ATPase domains, which is not really supported by the presented findings. It is therefore suggested to soften respective conclusions and modify the model accordingly.

6) The authors need to report on the refolding efficiency of aggregated Luciferase. So far only absolute activities (Luminescence a.u. Figure 1) are provided. Similarly, when comparing GFP unfolding or luciferase disaggregation activities of Hsp104 wild type and mutants, the scale of the y-axis is not well defined (Figure 3, Figure 5). It is recommended to set the activity of Hsp104 wild type at 100% and calculate the relative activity of variants.

*Reviewer #4:*

In their manuscript, Heuk et al. provide the structure of the Hsp104 disaggregase from *Chaetomium thermophilum* (a thermophilic fungus) at 3.7 A resolution. Subsequently, they carry out very detailed and careful biochemical analyses to highlight the mechanism by which the coiled-coil domain in Hsp104 regulates the activity of the chaperone. They also carry out mutational analyses to demonstrate how the coupling between the AAA1 and AAA2 ATPase domains is achieved in this protein.

As a general comment, the data presented in this manuscript about Hsp104 structure and mechanism of function are already established for Hsp104 and ClpB from other organisms. The only new information is probably the potential coupling of the activity of AAA1 and AAA2 through the pre-sensor 1 hairpin (PS1-hairpin). Furthermore, the interpretation of the data in several instances seems to be over simplified.

1) The authors need to explain Figure 1 and not just refer to it in the text.

2) The XL-MS data of Figure 4 might be misleading. The data is simply interpreted by the authors as: if there are more crosslinks, then the protein is more dynamic, which is not necessarily true. Furthermore, in Figure 4—figure supplement 2, the authors state that they needed to use larger amounts of BS3 to crosslink the hyperactive mutant compared to WT. This makes interpreting the crosslinking data problematic.

Also, what about crosslinks to the N-domain? These were not discussed. What happens if the crosslinking was done in the presence of ATP/ADP/etc.?

3) Figure 5 – It is not clear why Hsp104 with BMOE crosslink would be more active than WT?

4) To further detail the coupling between AAA1 and AAA2, the authors make several mutants including PS1+ and PS1- (making the PS1 hairpin longer or shorter). However, since such mutations can lead to many other structural re-arrangements, I think the authors have to be more qualitative in interpreting their data. The best case scenario would have been if the authors obtained the X-ray structures of these mutants, which I acknowledge might or might not be trivial.

[Editors’ note: what now follows is the decision letter after the authors submitted for further consideration.]

Thank you for resubmitting your work entitled "Structural basis for the disaggregase activity and regulation of Hsp104" for further consideration at *eLife*. Your revised article has been favorably evaluated by Michael Marletta (Senior Editor), a Reviewing Editor, and three reviewers.

The manuscript has been improved but there are some remaining issues that need to be addressed before acceptance, as outlined below:

One outstanding question that should be discussed before final acceptance is whether a helical assembly as described in a recent paper by Yokom et al., 2016 can be physiological. Is the mechanical link coupling the two AAA rings and the role of the middle domain compatible with a helical assembly as the functional disaggregase? A short critical comparison of the mechanistic features as revealed by the ADP-bound crystal structure (helical filament) of the Hsp104 from *Chaetomium thermophilum* with that of the recently published ATP-bound cryo-EM spiral structure of Hsp104 from yeast (Yokom et al., 2016), would be important for understanding how some AAA enzymes function.

Note that the mutational analysis requested by reviewer #1 are not absolutely required for final acceptance. However, the term CCD should be changed to M-domain throughout the manuscript and figures. Reviewer #2 is correct that the term M-domain is the standard used in the literature and it is not necessary to reinvent new terms.

*Reviewer #1:*

The authors addressed many of the comments made by mainly counter arguing but they also did carry out a few additional experiments. Given the recent publication of the cryo-EM structure of *Saccharomyces cerevisiae* Hsp104 in the ATP state by Yokom et al. [Nat Struct Mol Biol 23(9), 830-837], I think the publication of the current manuscript would be very timely.

*Reviewer #2 (General assessment and major comments (Required)):*

In their revised version the authors have successfully addressed most of my previous concerns. They added new data, which provide further support for the suggested role of the PS1 hairpin in signal transduction between the two ATPase domains and at the same time soften the model as formerly requested.

*Reviewer #3:*

The manuscript by Heuck et al. describes the 3.7A resolution crystal structure of a fungal Hsp104 together with biochemical and cross-linking/mass spectrometry (XL-MS) data. The manuscript is well written and experimental results are convincing. Perhaps one of the most interesting finding is that the CCD via motif-2 contacts the AAA1 large subunit of the neighboring protomer providing the structural basis for a functional role of the CCD in nucleotide signaling between neighboring ATPase modules. The latter has largely been inferred but never been demonstrated for an Hsp100 chaperone. The role of the PS1 motif in coordinating the two ATPase rings is novel and supported by biochemical experiments. The hypothesis that the CCD belt immobilizes the entrapped AAA1 ATPase modules is novel and substantiated in part by the XL-MS analysis using BS3 crosslinker. The only concern with this approach is that, because of the homo-oligomeric structures, intra- vs. inter-domain crosslinks would be difficult to differentiate.

1) It is not entirely clear why heterotypic contacts cannot be resolved by the recent cryoEM analysis of a proposed helical Hsp104 assembly (Yokom et al., 2016). Although it remains uncertain whether a helical assembly is physiological, is the proposed role of the PS1 motif compatible with a helical assembly?

---

## [Author Response]

[Editors’ note: the author responses to the first round of peer review follow.]

*The referees expressed concern regarding the quality of the structural model (reviewer #1), interpretation of the crosslink data (reviewers #2 & #4), controls for various biochemical experiments (reviewer #3) and the correct citation of the literature (reviewer #2).*

*We hope you will find the referee comments useful in preparing the manuscript for resubmission to eLife or another journal.*

*Reviewer #1:*

*Hsp104 is the fungal homolog of the bacterial ClpB and ClpC proteins. These hexameric molecular machines consist of two AAA (AAA1 and AAA2) rings. A characteristic feature is a coiled-coil insertion (CCD) in the AAA1 ring. The α-helical propeller-like structure of the CCD is important for functional cooperation with Hsp70 in protein disaggregation. In fungi, Hsp104 is a crucial component of the response to proteotoxic heat stress. Extensive structural and mechanistic studies of this class of molecular chaperones have previously been published. Crystal and cryo-EM structures exist for ClpB/ClpC and Hsp104, respectively. Numerous mutants, especially relating to the function of the CCD, have been characterized. However, questions about the regulation and the mechanism remained unanswered.*

*The present manuscript reports the first crystal structure of the eukaryotic disaggregase protein Hsp104. The crystal lattice does not contain the functionally active hexamer, but rather a helical filament of subunits. However, the structural building block, a complex between large and small AAA subdomains, resembles closely the respective units in the hexamer structure of ClpC.*

*I am surprised about the poor stereochemical quality of the model, with 82.6, 13.3 and 4.1% of the peptide bonds in favored, allowed and disallowed regions of the Ramachandran plots, respectively. For an excellent-quality model >98, <2 and <0.05% are expected. This suggests serious over-fitting of the data, thereby distorting the stereochemistry, and thus casts doubt on the atomic presentation/discussion of the biochemically important contacts. Restraining the stereochemistry more tightly and accepting slightly worse R-factors should yield a more plausible model. I also wonder whether the nominal resolution was properly chosen. The authors should consider using available automated tools like Rosetta_refine to improve the refinement of their model.*

In judging the quality of the current structure, one has to keep in mind the medium resolution of about 3.7 Å. At this resolution, the relatively "spacious" electron density does not define the precise geometry of the amino acid building blocks. To overcome this limitation, the information from high-resolution, homologous structures should be included in the refinement process. Unfortunately, however, for Hsp104, there are no suitable reference structures available that could be used as templates for the AAA1 or AAA2 domain. The only unfoldase present in a functional oligomeric state is ClpC, but the disaggregase is too distantly related to Hsp104 and was also determined at medium resolution (3.65 Å). Similarly, the published structures of ClpB, in which the AAA domains were captured either in isolation or as distorted ATPase rigid bodies, did not serve as suitable reference models. When used in DEN or Phenix refinements, the ClpC- and ClpB-derived models improved the stereochemical properties of Hsp104, but pulled the structure out of the density, as reflected also by increasing R-free values. With regards to Rosetta, it should be noted that the suggested Rosetta_refine tool does not properly work for large proteins, such as Hsp104, having ≥ 1000 residues. Even more importantly, it is not possible so far to impose NCS restraints in Rosetta_Refine. However, applying NCS restraints was absolutely required in the present case to ensure the continuous and parallel decrease of R and R-free. Moreover, we had to activate the "Ramachandran" option in Phenix_Refine (Headd et al., 2012) to enhance the stereochemical properties during refinement. To further improve the overall quality of the reported model, as recommended by this reviewer, we manually corrected the Hsp104 structure in iterative rebuilding and refinement cycles. The final Ramachandran statistics (favored: 86.8%, allowed: 12.9% , outlier: 0.2%, in comparison to 82.6% , 13.3% and 4.1% of the original model) and stereochemical parameters (RMSD's for bonds: 0.006 and angles: 1.08) nicely compare to other crystal structures determined at similar resolution and lacking a homology model. The improvement of the model is manifested in the drop of the R-free value from 0.294 to 0.277, as shown in the new crystallographic Table (Figure 1—figure supplement 1). Together, the crystallographic parameters reflect a well-refined crystal structure at 3.7 Å resolution and exclude an over-fitting of the diffraction data. The updated coordinates are now submitted to the Protein Data Bank replacing the previous entry (PDB 5d4w).

*The biochemical evidence presented for a CCD belt being a "straitjacket" for the AAA1 domain mobility appears solid. However, it is strange that the reduced Cys-Cys mutant of Hsp104 (Figure 5) should yield much lower activity than wildtype and the BMOE-linked variant. The proposed model appears to be in line with previously published biochemical data.*

The introduced Cys residues Q491C/G380C, located at the CCD-AAA1 interface, are in close vicinity to each other such that they can form a disulfide bridge under oxidizing conditions. Owing to this close proximity, we presume that under reducing conditions the two cysteines form a short-distanced hydrogen bond that stabilizes Hsp104 in its repressed state (of note, the original Gly-Gln pair does not interact with each other). In contrast, insertion of the bulky BMOE crosslinker should physically separate the two cysteine residues, yielding an activated state. To explore the influence of the two cysteines on Hsp104 activity, we modified Cys380 and Cys491 with maleimide, the functional group of the BMOE cross- linker. According to the Hsp104 crystal structure, maleimide attachment should also disrupt the Cys-Cys interaction and activate Hsp104. Consistently we observed that maleimide treatment elevated Hsp104 activity to a similar extent as the BMOE cross-linker. In conclusion, the new data explain the different activities of the Q491C/G380C variants (red, ox, BMOE, Mal). Most importantly, the two cross-linked particles, Cys‒Cys and Cys‒BMOE‒Cys, exhibit distinct activities. While both cross-linkers tether and thus physically restrain adjacent ATPase units, only the BMOE compound provides the conformational freedom to let the engaged ATPase units move against each other and remodel client proteins. The new data are included in the following paragraph (Figure 5):

“To estimate the effects of “tight” and “loose” CCD belts, we carried out Cys-Cys and Cys- bismaleimidoethane-Cys (BMOE) cross-linking, respectively, and compared the activities to those under no-cross-linking conditions. […] In conclusion, the cross-linking data support the restrain-mask model showing that AAA1 domains engaged by a covalently-linked but loosened CCD belt can still reorient and cooperate with each other.”

Reviewer #2: […]

*On balance, the data support the conclusion drawn. Although the work is a welcome addition and may be suitable for publication, is requires major revisions. Especially care should be taken when revising the manuscript because some of the references are incorrect or misleading, while others are missing. A re-review of the revised manuscript is advisable due to the collection of minor issues that distract from the main thrust of the current paper.*

*1) It is unfortunate that the crystal structure of CtHsp104 and not of ScHsp104 was determined, which does not directly address the controversy that has been damaging the field. For instance, the biochemical activities of CtHsp104, while similar to ScHsp104, are not identical, and could fuel the argument that CtHsp104 is different from ScHsp104. Is CtHsp104 essential for fungal prion propagation?*

The activities of the two Hsp104s differ to some extent in the applied disaggregation assay. Of note, the activity measured in this assay also depends on the cognate Hsp70 that varies between Ct and Sc, a fact that also prevents testing the role of CtHsp104 for prion propagation in yeast. Most importantly, however, the molecular mechanisms behind rescuing protein aggregates should be conserved between CtHsp104 and ScHsp104. To show this biochemically, we included a ScHsp104 variant lacking the PS1 hairpin in our analysis (Figure 6—figure supplement 1). ATPase and GFP unfoldase data suggest that the yeast enzyme utilizes the same AAA1-AAA2 coupling mechanism as CtHsp104, to remodel damaged proteins. We thus presume that findings for the CtHsp104 system can be straightforwardly extrapolated to related HSP100 disaggregases. These data are referred in the revised text:

“Notably, deleting the PS1 hairpin in the yeast Hsp104 had a similar effect, suggesting that the uncovered coupling is generally relevant for HSP100 disaggregases (Figure 6—figure supplement 1).”

*2) A major limitation of the present work is that the X-ray structure of the ring assembly was not determined. Moreover, neither the N-domain nor C-terminal extension were resolved in the present structure, perhaps as a result of finding a structure solution by molecular replacement, arguing for independent phase information.*

Not having captured the hexamer is clearly not our favorite scenario. However, we would like to emphasize again that the basic building blocks of the hexamer are maintained in the observed filament. Thus, we could deduce important mechanistic features from the structural data, which we later confirmed by a detailed biochemical analysis.

With regards to model building, the reviewer is right – the protein was crystallized as SeMet protein thus offering independent SAD phase information. Although this phase information was very helpful at the initial stages to interpret the AAA1, CCD and AAA2 folds, the NTD and the C-terminal extension did not show up in the experimental density maps. However, this is not surprising. While the C-terminal extension is predicted to be disordered, the NTD could not (or only weakly) be seen in the non-averaged electron density maps of recent cryoEM analyses (Lee et al., 2007). Thus, the NTD appears to exhibit inherent en- bloc mobility hindering its localization in a functional disaggregase and determining its precise role. In the present crystal structure, we observed extra, unconnected density in the vicinity of the CCD. However, the quality of this density was not good enough to include a model there. Figure 9 depicts part of the 2Fo-Fc electron density map next to molecule B, which showed the largest amount of extra electron density compared to the other protomers in the asymmetric unit:

Author response image 1.Extra density that may account for the N-terminal domain.Cartoon representation showing the CCD of molecule B colored in green, together with the 2FoFc electron density contoured at 0.8 σ. The observed extra density for molecule B that could not be used for model building is highlighted. The neighboring subunits (molecule C and crystallographic neighbors) are colored in grey.**DOI:**
http://dx.doi.org/10.7554/eLife.21516.029

*3) The authors do not make it clear how many of the cross-linked Lys, if any, disagree with their proposed model of active and inactive states. It would seem that interpretation of the cross-linking pattern is complicated since the structure of the active state was not determined. If the CCD represents a "restraint mask" or negative regulator of Hsp104, elimination of the CCD should activate the protein disaggregase. Yet, Hsp104 variants that lack the CCD are inactive in protein disaggregation. How can this be reconciled?*

Active and inactive states are characterized by distinct motilities of the same rigid bodies, formed between adjacent subunit. In other words, we expected to find all cross-links formed in the repressed state also in the wild-type and the active state. This was indeed the case, with wild-type and activated Hsp104 having increasing numbers of additionally connected Lys-Lys pairs, respectively. In total, 65% of the contacts that we observe for the repressed state are formed between lysine residues that are located at a distance of 10-30 Å. Thus, the observed MS pattern is consistent with and further corroborates our structural data. As noted by reviewer 4, several XL-MS contacts, however, do not fit to the modeled hexamer. For example, the flexible half of the CCD (motif-1) undergoes long-distance contacts with surface residues on the AAA2 domain. Such contacts, most of which were observed in the hyperactive Hsp104 variant, could result from the induced flexibility of motif-1 upon opening the CCD ring. Consistent with this interpretation, motif-2 of the CCD was also too flexible to be properly built into EM density maps of the hyperactive ClpB variant (Carroni et al., 2014). Please see also our comments to reviewer 4, major comment 2.

As suggested by the reviewer, we tested the activity of the CCD deletion mutant (CtHsp104ΔP427-G553). In our hands, the ΔCCD mutant showed elevated ATPase activity in the presence of substrate and an elevated GFP unfolding activity. However, consistent with the results from previous studies (Sielaff and Tsai, 2010), we observe a strongly reduced disaggregation activity (see Figure 10). Because the disaggregation activity depends on the binding of Hsp70 to the CCD, such reduced activity is not surprising and does not argue against the presented model. Moreover, it is not unlikely that the deletion of the entire CCD domain (residues 427 to 553) leads to further structural changes in the two-ring disaggregase. Therefore, point mutations that specifically disrupt CCD-AAA1 or CCD-CCD* contacts are better suited to address the regulatory function of the HSP100 coiled-coil domain.

Author response image 2.Activity of the CCD deletion mutant.Substrate-induced ATPase activity of CtHsp104ΔP427-G553 (ΔCCD) is higher than for wild-type Hsp104. Moreover, the ΔCCD mutant showed an increased GFP-unfolding activity while the luciferase disaggregation activity of the corresponding Hsp104/Hsp70 system was reduced.**DOI:**
http://dx.doi.org/10.7554/eLife.21516.030

*4) Care needs to be taken when reporting WA and/or WB Hsp104 mutants. It is known that WA mutants cause deoligomerization, while WB mutants can bind ADP or contain no nucleotide. At a minimum, the authors need to show that the mutants form hexamers.*

We now included the respective SEC profiles of CtHsp104 WA and WB mutants showing their structural integrity (Figure 6—figure supplement 2). Regarding the possible binding of ADP to the WB mutant, it is important to note that our biochemical assays contained relatively high ATP concentrations (5 mM) that exceed the observed dissociation constants of 12 μM for AAA1 and 0.1 μM for AAA2, respectively. Moreover, all assays were performed in the presence of an ATP regenerating system to shift the equilibrium to the ATP bound state. We thus believe that under the experimental conditions used, the WB mutants should represent the ATP-bound state.

*5) The novelty of the PS1-hairpin of AAA-2 is somewhat overstated since it is identical to the previously reported β-hairpin that is part of the intersubunit signaling motif (Biter et al., 2012).*

The mentioned study describes a structural characterization of the isolated AAA2 domain from *thermophilus* ClpB. In this structure, the authors observe a direct contact between the PS1 hairpin residue H693 (Q732 in CtHsp104) and the AAA2 pore loop. Based on this observation, the authors suggest an intra-ring signaling mechanism that involves the residues D685 (D724 in CtHsp104) and R747 (Arg-Finger, R788 in CtHsp104). Although H693 is not conserved within the class of HSP100 proteins ([Supplementary-material SD3-data]) and contacts between the PS1 hairpin and the pore loop are not visible in our CtHsp104 crystal structure, the observed contact may reflect an alternative conformation and function of the PS1 hairpin. Indeed, the connection of PS1 and the pore-loop of AAA2 is interesting, as it may point to a way of coupling inter-ring structural changes with the rearrangement of the pore-loops in AAA2. We have to apologize for not having included the indicated mechanistic study in our Discussion. At the same time, however, we do not feel that the novelty of the PS1 hairpin is overstated, as our study elucidates the long-sought structural motif coupling the activities of the two AAA rings in HSP100 disaggregase machines. To properly account for the work of Biter et al., we have added the following statement to our manuscript:

"Aside from the inter-ring communication, the PS1 loop has been implicated in intra-ring signaling, l to its location near the pore-loop of AAA2 (Biter et al., 2012b).”

*Additional comments:*

*1. It is now generally accepted that the Wendler model (Wendler et al., 2007) is wrong, falling short of a retraction. It is therefore confusing what the authors consider "consistent" with the Wendler model (e.g. subsection “Crystal structure of the Hsp104 subunit”, last paragraph). Moreover, it is doubtful that the Wendler structure has the necessary resolution to assign side-chain conformations correctly. Great care needs to be taken when using the terms "flexible", "motions", "structural rearrangements, which are loosely defined and refer to very different things in high-resolution X-ray structures and in structural work based lower resolution cryoEM, SAXS, etc.*

We have revised our manuscript to avoid non-proper use of the indicated terms. We also removed the (Wendler et al., 2007) citation from the indicated statement. However, one mechanistically relevant arginine located on the CCD has been identified in this study (Arg509), although for an entirely different (as it looks now, wrong) reason. Arg509 was implicated in nucleotide binding, whereas the present structure demonstrates that mutating this arginine should detach the CCD from the AAA core, thus unrestraining and activating the ATPase machinery (Figure 3).

*2) Introduction: "Mixed L/S* modules…represent the functional units of AAA chaperones[…]". The large and small subdomains are both required for nucleotide binding. Neighboring subdomains may exert regulatory function but are dispensable for nucleotide binding in cis. This statement is misleading and needs to be revised.*

We agree. The revised statement reads now as follows:

"Substrate stretching and unfolding is mediated by ATP-driven power strokes (Maillard et al., 2011), which result from the movement of rigid ATPase bodies composed of the large (L) subdomain of one protomer and the small (S) subdomain of the next (Glynn et al., 2009, Wang et al., 2001). Coordination of adjacent L/S* modules (the asterisk denotes the neighboring subunit) relies on a special active site organization, as each nucleotide binding site is formed by residues of the L-, S*- and L*-subdomains at the subunit interface."

*3) Abstract: What is exactly meant by "near-atomic"? Does it refer to resolution (i.e. 3.7A) or coordinates (i.e. atoms can be seen)? Near-atomic is not appropriate to describe either (c.f. Results, first paragraph).*

We have replaced near-atomic by the nominal resolution of the determined crystal structure:

“we present structural and biochemical data revealing the organization of the Hsp104 disaggregase machinery at 3.7 Å resolution”.

*4) Introduction, first paragraph: It is true that AAA ATPases are found in cells of all organisms, but Hsp104 disaggregases are not present in animal cells, despite functional conservation of a protein disaggregating activity in higher Eukarya. This needs to be clearly distinguished.*

We have changed the above statement to:

"HSP100 unfoldases, a subclass of the AAA chaperones found in yeast and bacteria as well as in the mitochondria and chloroplasts of higher eukaryotes, employ a powerful mechanism to reactivate damaged proteins."

*5) Introduction, second paragraph: The present work is concerned with the structure-function analysis of C. thermophilum Hsp104. CtHsp104 and ScHsp104 are similar but non-identical (e.g. Figure 1). Also, in Figure 1 (right panel), it is unclear whether the luminescence measurements represent absolute or relative activities. What is the activity of the cognate ScHsp104:Hsp70/40 system? What is the SEM?*

In the original Figure 1, absolute numbers of luciferase activity were presented. However, to highlight differences between independent experiments we now present activities normalized to wild type CtHsp104 ( ± standard deviation). As indicated before, the Hsp104/Hsp70/Hsp40 systems from Sc and Ct exhibit distinct activities in rescuing heat-denatured luciferase. However, most importantly, the applied mechanism for remodeling aberrant proteins should be conserved, as suggested by the analysis of the ∆PS1 mutant of ScHsp104 (please see also major-point 1).

*6) Introduction, second paragraph: The ability of Hsp104 to unravel prions on its own has been questioned (Inoue et al., 2004; Krzewska and Melki, 2006; Reidy et al., 2012; Yuan et al., 2014). It is now widely accepted that the ability to dissolute prions is dependent on the Hsp104-Hsp70 bi-chaperone system, and is the same for the dissolution of for amorphous aggregates.*

We agree and have changed the text accordingly.

"Indeed, Hsp104 can team up with Hsp70 to establish one of the most potent disaggregase machineries in nature, being able to unravel even the particularly resistant amyloid fibers (Shorter and Lindquist, 2004, Inoue et al., 2004, Krzewska and Melki, 2006)."

*7) Introduction, second paragraph: Rosenzweig et al. did not demonstrate that Hsp70 binding to the M-domain activates the Hsp100 motor (Rosenzweig et al., 2013). They showed that DnaK binds to the M-domain of Hsp100 confirming earlier work by Haslberger et al. (Haslberger et al., 2007).*

The indicated sentence “Binding of the Hsp70 chaperone to the CCD activates Hsp104 and targets it towards protein aggregates (Lee et al., 2013, Seyffer et al., 2012, Rosenzweig et al., 2013, Oguchi et al., 2012, Haslberger et al., 2007, Miot et al., 2011, Sielaff and Tsai, 2010).” refers to the interaction site of Hsp70 and the activation of Hsp104 upon Hsp70 binding. As Rosenzweig et al., 2013 visualized the DnaK binding to this domain, we feel that this citation is relevant at this point. However, we agree that the study by Haslberger et al., 2007, Miot et al., 2011 should be cited here as well.

*8) Introduction, end of second paragraph: It is now widely accepted that the Wendler structure (Wendler et al., 2007) is incorrect and contrasts the more recent work by the same authors (Carroni et al., 2014). This statement needs to be revised. In this context, recent work by the Bukau, Wickner and other labs how the CCD regulates the Hsp100 motor should be discussed.*

We have adapted the text accordingly. Regarding the latter point, a detailed introduction describing the regulatory function of the CCD is already provided and should give the broader audience the required background on the HSP100 coiled-coil insertion:

“Recent studies highlight the central role of the CCD in regulating HSP100 disaggregases, as the domain is critical for AAA1-AAA2 communication, Hsp70 binding, and keeping the enzyme inactive in the absence of cognate substrates (Cashikar et al., 2002, Lee et al., 2013, Oguchi et al., 2012, Seyffer et al., 2012, Haslberger et al., 2007, Sielaff and Tsai, 2010, Miot et al., 2011).”

*9) Subsection “Crystal structure of the Hsp104 subunit”, last paragraph: How was the identity of sensor-2 defined? There should only be one, not three.*

According to the definition of (Hanson and Whiteheart, 2005),Arg402 and Arg849 are the sensor-2 residues of AAA1 and AAA2, respectively, located at the beginning of the conserved AAA helix-α7. This helix protrudes towards the bound nucleotide and carries a number of highly conserved active site residues. Owing to the crucial regulatory role of these residues, helix-α7 is often referred to as sensor-2 helix. We would thus prefer to keep the used nomenclature, which, also, was not criticized by the other reviewers. To avoid any misunderstanding, we added the indicated reference:

"In AAA1, ADP is accommodated in a pocket formed by the general Walker A (Lys229, Thr230) and Walker B (Asp295, Glu296) motifs as well as by AAA-specific (sensor-1: Thr330, sensor-2: Arg402) functional groups (Hanson and Whiteheart, 2005, Mogk et al., 2003, Hattendorf and Lindquist, 2002b)."

*10) Subsection “Crystal structure of the Hsp104 subunit”, last paragraph: "[…]Arg349 has been previously described as the Arg finger[…]? What is the reference for this?*

The reference is (Mogk et al., 2003) and we included the citation in the manuscript (subsection “Crystal structure of the Hsp104 subunit”, second paragraph).

*11) Subsection “ATPase rigid bodies are maintained in the crystallized Hsp104 filament”, last paragraph: Glynn et al. (Glynn et al., 2009) reported the hexameric crystal structure of ClpX, a single-ring Hsp100 member. This statement needs to be rephrased. Also, what about ring-forming structures determined by cryoEM?*

We have rephrased the sentence accordingly, referring in particular to HSP100 enzymes containing two AAA rings. Due to the lower resolution of the available cryo-EM structures, respective models were not considered for comparison with the Hsp104 crystal structure:

“Strikingly, structural comparison with the related AAA protein ClpC (Wang et al., 2011) – the only double-ring HSP100 unfoldase crystallized in a hexameric form so far – shows that the AAA1L/AAA1S* and AAA2L/AAA2S* rigid bodies of the Hsp104 filament are very similar to those of the hexameric unfoldase (Figure 2).”

*12) [Supplementary-material SD3-data]: The sequence numbering does not match with the sequence numbering of the atomic coordinates.*

We have checked the numbering again, and, in our opinion, the sequence numbering is matching the atomic coordinates.

*Reviewer #3:*

*AAA proteins are mechanochemical ATPases that perform conformational work fueled by ATP hydrolysis to remodel substrates. The yeast AAA protein Hsp104 functions as a disaggregase by reactivating aggregated proteins in cooperation with Hsp70. Hsp104 consists of two AAA domains (AAA-1, AAA-2) and a regulatory coiled-coil domain (CCD, also termed M-domain). How ATPase activity is regulated and how the two ATPase modules communicate is of central importance for understanding disaggregase mechanism. Here the authors determined the crystal structure of Chaetomium thermophilum Hsp104 providing them a structural map to dissect the regulation of ATPase and disaggregation activity.*

*The determined crystal structure is similar to the ones of bacterial ClpB homologs. However, it includes additional novel and valuable information as crucial contacts between neighboring AAA domains are maintained in contrast to former structures. The novel Hsp104 structure confirms the previously established position and regulatory function of the CCD yet the interaction details are of higher resolution. The authors show that the CCD is restricting AAA-1 mobility, providing a rationale for down regulation of ATPase and disaggregase activity by the CCD. This part of the study is well performed and includes novel information, yet it also represents an evolution of an already established and accepted mode of Hsp104 activity control. In the second part, the authors identify a conserved structural element, the PS1 hairpin, which is suggested to act as sensor mediating interdomain communication between AAA-1 and AAA-2. A role of the hairpin in controlling ATPase and disaggregase activity is supported by biochemical analysis, though some assays and results need further clarification (see below). How exactly PS1 synchronizes the AAA modules remains vague and therefore some of the conclusions should be softened.*

*Overall the presented work is of high quality, contributes to an improved understanding of disaggregase mechanism and includes the identification of a novel regulatory element. The following points should be addressed in a revised manuscript:*

*1) The RepA-GFP unfolding assay requires the presence of non-physiological ATP/ATPγS mixtures. The relevance of such conditions for disaggregase function (disaggregation activity in presence of these nucleotide mixtures is low or not existing) and ATPase communication is unclear and results based on the assay are at least in parts questionable.*

We agree that the ATP/ATPγS mixture in the RepA-GFP unfolding assay is not optimal. However, the assay is well-established in the field. The nucleotide mixture was introduced, because ATP is required for binding substrates, whereas ATP hydrolysis is critical for the remodeling of the captured substrates. For protein disaggregation, the situation is different as substrate binding is mediated by the Hsp70 partner chaperone. Therefore, the disaggregation assays can be carried out under more physiological conditions lacking ATPγS. Accordingly, the two assays, when applied in combination, should represent a proper way to characterize HSP100 variants and determine the kinetic parameters of the unfolding as well as the disaggregation reaction. Consistent with this, most mutant proteins that were predicted to affect the communication between AAA1 and AAA2 ATPase rings were similarly affected in their unfoldase and disaggregase activity. To further validate the proposed mechanism, we have now also performed ATPase measurements characterizing the communication of AAA1 and AAA2 in the presence of substrate (see next point). Together, the unfoldase, disaggregase and ATPase data provide strong evidence for the role of the PS1 hairpin in coupling AAA1 and AAA2 ATPase activity.

*2) ATPase activities were always determined in the absence of substrate. Casein and specific peptides stimulate ATPase hydrolysis by Hsp104. It is recommended to include those in ATPase measurements as it might allow to more precisely defining the role of PS1 in signal transduction.*

As suggested, we carried out ATPase assays (i.e. for the mixed walkerA/B mutants) in the presence of casein, our model substrate (Figure 6—figure supplement 3). Consistent with the numbers observed in the absence of substrate, the derived kinetic parameters nicely confirm the coupling of the two AAA engines, as is now described:

"Of note, the same effect was observed in the presence of substrate proteins, emphasizing the role of the PS1 hairpin in coupling the two AAA engines during substrate translocation (Figure 6 and Figure 6—figure supplement 3).”

*3) To exclude that deletion of the PS1 hairpin causes structural defects it is recommended to analyze the effect of e.g. Q732/R734 mutation, which should abolish interaction with the sensor 2 helix.*

We followed the advice of this referee to introduce an additional point mutation into the PS1 hairpin. As both residues Q732 and R734 form various hydrogen-bonds at the AAA1 active site and because the QR/AA double mutant could not be produced as stable protein, we decided to mutate another residue at the tip of the PS1 hairpin. By replacing Gly731 we aimed to abrogate the close contact with Trp573 (Figure 6) and sterically disrupt the binding of PS1 to AAA1. The corresponding G731R mutant had an overall reduced ATPase activity, however its unfolding and disaggregase activities were even more impaired, thus supporting the results of deleting the PS1-hairpin. To this end, we would like to note that the WA/WB mutants lacking the PS1 hairpin exhibited robust, substrate-inducible ATPase activity in both AAA rings (Figure 6 and Figure 6—figure supplement 3). These data indicate that the intra-ring communication is still active, excluding gross structural defects by the PS1 deletion. Moreover, we characterized further site-specific mutants in the PS1 motif (PS1+, PS1- (Figure 7 and Figure 7—figure supplement 2) that exhibit a similar phenotype as the ∆PS1 mutant. Together, the biochemical characterization of the PS1 variants supports the proposed role of PS1 in mechanically coupling AAA1 and AAA2:

"To show that the observed effects are not due to putative gross structural changes caused by the PS1 deletion, we analyzed a site-specific mutation at the tip of the PS1 hairpin that was predicted to sterically expel the PS1 motif from the active site of AAA1. For this purpose, we replaced Gly731, which is in close contact to Trp573, by arginine. When tested in our activity assays, the G731R mutant had a slightly decreased ATPase activity, but it was even more impaired in its unfoldase and disaggregase activity, thus mimicking the PS1 deletion phenotype (Figure 6).”

*4) Figure 7: The two analyzed variants of N748 (sensor1) have opposing effects on GFP unfolding (Figure 4), a conflict overlooked and not discussed so far.*

Indeed, the N748S and N748Q mutations have opposite effects on the unfolding potential of Hsp104. Since both mutations completely suppress AAA2 activity, we cannot distinguish whether the catalytic activities observed for them stem from a local function of sensor-1 in the nucleotide binding pocket of AAA2 or rather point to a role in the allosteric communicating between different parts of AAA2 or between the two AAA rings. At the current stage, we find the results of the unfolding and disaggregase assays difficult to explain in definite terms. For that reason, we decided to omit these data from Figure 7, but still share them with the community in the supplementary part of the manuscript (updated Figure 7—figure supplement 3).

Instead, we focused on the ATP binding assays, which are easier to interpret. Both mutations of the sensor-1 residue in AAA2 markedly increase the binding affinity of AAA1 for ATP (Figure 7) demonstrating that slight changes in sensor-1 of AAA2 are communicated to the remote AAA1 ring. These results, together with the new observed data for the PP mutant (Figure 7) support our hypothesis that the sensor-1 residue and the central β-strand (β4) are part of the PS1 signaling device synchronizing the activities of the two AAA rings.

The opposite effects mediated by the sensor-1 mutations are now described in the manuscript: “Although mutating the sensor-1 residue of AAA2 had only a minor influence on the overall ATPase activity (Figure 7), the two mutations clearly affected the nucleotide binding in the remote AAA1 ring. […] Owing to the close distance of the PS1 hairpin and the AAA2 pore-loop (Biter et al., 2012b), we presume that the opposite activities of the sensor-1 mutants may reflect different substrate translocation properties of the Hsp104 particles; however, the molecular mechanism of this intriguing function remains to be elucidated.”

*5) While the data demonstrate that PS1 plays a role in ATPase communication, the precise mechanism remains unclear. In their model (Figure 8) the authors suggest a defined order of signaling events between the ATPase domains, which is not really supported by the presented findings. It is therefore suggested to soften respective conclusions and modify the model accordingly.*

We agree and adapted the Discussion and Figure 8 accordingly. Moreover, we characterized a further mutant supporting the proposed signaling mechanism. An important feature of the HSP100 machinery is the distorted β-sheet in AAA2 allowing for rearrangements of individual β-strands in response to external stimuli. As indicated by the crystal structure, strand β4, the direct extension of the PS1-hairpin, is only loosely connected to its neighboring β-strands. The distortion in secondary structure is mainly due to a highly conserved Pro-Pro motif in the adjacent β1 strand. To more strongly fasten together strands β1 and β4, we replaced the Pro627/Pro628/Ser629 by the Thr627/Gly628/Asn629 motif, a sequence found in Hsp104 homologues from diatoms, such as Coscinodiscophyceae or Bacillariophyceae. Consistent with our model, this mutant exhibited an increased ATPase activity; however, it was largely inactive in GFP- unfolding and luciferase disaggregation assays (Figure 7). Moreover, the AAA2 ATPase activity of this mutant was uncoupled from the nucleotide binding state of AAA1 (Figure 7—figure supplement 2). Accordingly, the P627T-P628G-S629N mutant exhibits a similar mechanistic phenotype as the PS1 deletion mutant.

Taken together, we present a detailed characterization of 5 mutants (PS1+, PS1-, PP, G731R and ∆PS1) that abrogate the AAA1-AAA2 communication at distinct strategic points. Importantly, all these mutants show a similar enzymatic profile in retaining ATPase activity, but being incapable of remodeling aberrant proteins. Together, these data confirm the proposed role of the PS1 motif in coupling AAA1 and AAA2 activities, as is now described:

"We suggest that the PS1-hairpin and the associated strand β4 couple conformational changes in AAA1 with the repositioning of the catalytic sensor-1 residue of AAA2 (Figure 8). […] This may be one of the factors underpinning the dynamic and highly allosteric nature of AAA proteins."

*6) The authors need to report on the refolding efficiency of aggregated Luciferase. So far only absolute activities (Luminescence a.u. Figure 1) are provided. Similarly, when comparing GFP unfolding or luciferase disaggregation activities of Hsp104 wild type and mutants, the scale of the y-axis is not well defined (Figure 3, Figure 5). It is recommended to set the activity of Hsp104 wild type at 100% and calculate the relative activity of variants.*

As suggested, we indicate the luciferase refolding efficiency (which was between 5 and 15% , as mentioned in the Materials and methods section, subsection “Luciferase disaggregation assay”). With respect to the axis label, all unfolding and disaggregation activities are shown relative to the respective wild type level (wt set to 100% ). We improved our figure presentations by changing the axis label to: “% wt activity”.

*Reviewer #4:*

*In their manuscript, Heuk et al. provide the structure of the Hsp104 disaggregase from Chaetomium thermophilum (a thermophilic fungus) at 3.7 A resolution. Subsequently, they carry out very detailed and careful biochemical analyses to highlight the mechanism by which the coiled-coil domain in Hsp104 regulates the activity of the chaperone. They also carry out mutational analyses to demonstrate how the coupling between the AAA1 and AAA2 ATPase domains is achieved in this protein.*

*As a general comment, the data presented in this manuscript about Hsp104 structure and mechanism of function are already established for Hsp104 and ClpB from other organisms. The only new information is probably the potential coupling of the activity of AAA1 and AAA2 through the pre-sensor 1 hairpin (PS1-hairpin). Furthermore, the interpretation of the data in several instances seems to be over simplified.*

*1) The authors need to explain Figure 1 and not just refer to it in the text.*

Figure 1 is now introduced in the main text:

“we performed a biochemical and structural analysis of the Hsp104 disaggregase from *Chaetomium thermophilum*, which exhibits similar ATPase and protein remodeling activities to those of the well- characterized yeast ortholog (Figure 1).”

2) The XL-MS data of Figure 4 might be misleading. The data is simply interpreted by the authors as: if there are more crosslinks, then the protein is more dynamic, which is not necessarily true. Furthermore, in Figure 4—figure supplement 2, the authors state that they needed to use larger amounts of BS3 to crosslink the hyperactive mutant compared to WT. This makes interpreting the crosslinking data problematic.

*Also, what about crosslinks to the N-domain? These were not discussed. What happens if the crosslinking was done in the presence of ATP/ADP/etc.?*

We agree that the presentation of the XL-MS experiment may be misleading; however, our interpretation of these data is consistent with other studies in the field. As reported for example by (Walzthoeni et al., 2015, Scorsato et al., 2016), chemical crosslinking can be used in combination with quantitative mass spectrometry to visualize the dynamics of high-molecular weight complexes. The main difference to the indicated study is that we did not carry out a quantitative MS analysis of the cross-linked peptides (as this is highly complex requiring specialized software programs), but rather performed a semi-quantitative analysis comparing the number of cross-linked peptides. Such an analysis, i.e. correlating number of cross-links with dynamicity, has been also reported by others, as for example (Sriswasdi et al., 2014). In the case of Hsp104, we reasoned that in a rigid protein ensemble (e.g. the repressed state), the inter- residue distances are well-defined allowing efficient cross-linking of certain lysine residues, whereas in a dynamic protein complex (i.e. the hyperactive mutant) subunits move against each other yielding an ensemble of intermediate states. As the reaction with BS^[3]^ is irreversible, it is possible to capture the short- lived states thus obtaining a larger number of cross-links in a dynamic than in a rigid protein complex. Correlating the number of cross-links with the dynamics of the Hsp104 structure is also supported by the domain-specific effects induced by the hyperactive and repressive mutation. Whereas the number of cross- links varies within the AAA1 domain by a factor of 4, the differences in the AAA2 domain are markedly smaller (Figure 4—figure supplement 1). Accordingly, the dynamics of the AAA1 ring seems to be most affected by the CCD mutation, as predicted by the restraint-mask model.

Regarding the BS^3^ cross-linker, we indeed used different concentrations for the MS experiment. The reason for this is that we wanted to compare wild-type, repressed and hyperactive Hsp104 having a similar total number of cross-links. To identify respective conditions, we performed an SDS-PAGE analysis to characterize inter-subunit Hsp104 crosslinks in dependence of the BS^3^ concentration (Figure 4—figure supplement 1). From these experiments, we concluded that we should apply the concentration of 0.3 mM BS3 to cross link wild-type and repressed CtHsp104 and 0.6 mM BS3 for the hyperactive variant. In other words, the cross linking conditions for the wild type and the repressed Hsp104 variants were identical. To clarify the outline of the cross-linking experiment, we have modified the chapter in the Materials and methods section accordingly:

"The appropriate BS3 concentration (repressed variant and wild-type 0.3 mM, hyper-active variant 0.6 mM, respectively) was determined based on SDS-PAGE (Figure 4—figure supplement 1)."

Cross-links to the NTD are not discussed, as the location of the NTD is still unclear (the NTD is a highly mobile domain that could not be conclusively resolved in recent EM analyses). Likewise, we haven't analyzed the crosslinking pattern of different nucleotide-bounds states, as this would represent an analysis on its own going beyond the scope of this study.

For a better presentation of the cross linking data we updated Figure 4 and modified XL-MS paragraph in the Results section:

"To directly monitor the mobility of the engaged ATPase modules upon opening and closing of the CCD ring, we performed a cross-linking coupled mass spectrometry (XL-MS) experiment. […] Once the CCD contacts are broken, the ATPase modules are free to move against each other, as seen for the hyperactive mutant, to remodel engaged client proteins."

*3) Figure 5 – It is not clear why Hsp104 with BMOE crosslink would be more active than WT?*

To address this point, we treated Hsp104 with maleimide, the functional group of the BMOE crosslinker. Similar to the results observed upon BMOE crosslinking, also maleimide treatment stimulated Hsp104 activity above the wild type level, suggesting that the chemical modification of Cys380 and Cys491 increases Hsp104 activity, most likely by disrupting this 380-491 contact. More importantly, it should be noted that BMOE covalently cross-links two adjacent ATPase modules. It is thus remarkable that such physically restrained Hsp104 machine is still capable to be enzymatically active. Please refer also to our response to reviewer 1, major comment 2for further details. The additional experiments are now described in the following paragraph and Figure 5:

“To estimate the effects of “tight” and “loose” CCD belts, we carried out Cys-Cys and Cys- bismaleimidoethane-Cys (BMOE) cross-linking, respectively, and compared the activities to those under no-cross-linking conditions. […] In conclusion, the cross-linking data support the restrain-mask model showing that AAA1 domains engaged by a covalently-linked but loosened CCD belt can still reorient and cooperate with each other.”

*4) To further detail the coupling between AAA1 and AAA2, the authors make several mutants including PS1+ and PS1- (making the PS1 hairpin longer or shorter). However, since such mutations can lead to many other structural re-arrangements, I think the authors have to be more qualitative in interpreting their data. The best case scenario would have been if the authors obtained the X-ray structures of these mutants, which I acknowledge might or might not be trivial.*

We are aware that the insertion or deletion of individual PS1 residues may cause structural re-arrangements in the target protein. However, when analyzing these mutants, we observed a wild type-like ATPase activity in the two AAA rings. These data show the conservation of intra-ring signaling thus highlighting the overall structural integrity of the PS1 mutants (Figure 7 and Figure 7—figure supplement 2). In addition, we provide biochemical data of two additional mutants (G713R, Figure 6; P627T-P628G-S629N, Figure 7) that show comparable activity to the PS1+ and PS1- mutants while maintaining the length of the PS1 hairpin. As discussed previously (reviewer 3, major comment 5), the mutational analysis carried out during revision provides strong evidence for the proposed signaling mechanism, which is now presented in a more qualitative way. Certainly, structural data of the PS1 mutants is highly desirable, but unfortunately out of reach (finding well-diffracting crystals among the many weakly-diffracting crystals took about 3 years).

[Editors' note: the author responses to the re-review follow.]

*The manuscript has been improved but there are some remaining issues that need to be addressed before acceptance, as outlined below:*

*One outstanding question that should be discussed before final acceptance is whether a helical assembly as described in a recent paper by Yokom et al., 2016 can be physiological. Is the mechanical link coupling the two AAA rings and the role of the middle domain compatible with a helical assembly as the functional disaggregase? A short critical comparison of the mechanistic features as revealed by the ADP-bound crystal structure (helical filament) of the Hsp104 from Chaetomium thermophilum with that of the recently published ATP-bound cryo-EM spiral structure of Hsp104 from yeast (Yokom et al., 2016), would be important for understanding how some AAA enzymes function.*

*Note that the mutational analysis requested by reviewer #1 are not absolutely required for final acceptance. However, the term CCD should be changed to M-domain throughout the manuscript and figures. Reviewer #2 is correct that the term M-domain is the standard used in the literature and it is not necessary to reinvent new terms.*

To properly account for the recently published cryoEM structure of the Hsp104 from *Saccharomyces cerevisiae* (ScHsp104), we now compare the organization of the CtHsp104 filament with the two distinct hexameric states of HSP100 enzymes, which are represented by the helical ScHsp104 (Yokom et al., 2016) and the planar ClpC hexamer (Wang et al., 2011). We focused in particular on the question, whether the proposed function of the PS1-hairpin in mediating communication between the AAA1 and AAA2 rings is compatible with the distinct quaternary organizations. To this end, we aligned the L/S* rigid bodies observed in the CtHsp104 crystal structure en bloc onto corresponding building blocks of the ScHsp104 and ClpC hexamer. The resultant alignment clearly show that the PS1 motif is well positioned to functionally link the AAA1 and AAA2 rings in the both planar and helical arrangement of the hexamer. To include this comparison into the manuscript, we added the following paragraphs:

“In the crystal, CtHsp104 subunits are arranged in a helical 61 filament (Figure 1) rather than forming a defined hexameric particle as observed in the recent cry-electron microscopy structure of ScHsp104 (Yokom et al., 2016) or the crystal structure of a related HSP100 unfoldase, ClpC (Wang et al., 2011). […] In fact, structural comparison shows that the AAA1L/AAA1S* and AAA2L/AAA2S* rigid bodies of the Hsp104 filament are very similar to those of the hexameric Hsp104 and ClpC (Figure 2 and Figure 2—figure supplement 2).”

“To test whether the PS1 motif could adopt a similar conformation in the hexameric particle, we aligned the functional L/S* unit (AAA1L/S*-AAA2L/S*) of the CtHsp104 filament onto the ClpC and ScHsp104 hexamers. […] According to these data, we hypothesize that the Hsp104 disaggregase can switch between planar and helical conformations while maintaining the integrity of the L/S* rigid bodies to ensure intra- and inter-ring cooperativity during the ATPase-driven power strokes.”

To account for the inserted Figures (Figure 2—figure supplement 1, Figure 6—figure supplement 1), we had to revise the order of our supplement figures in the following way:

Figure 2—figure supplement 1: Comparison between planar and helical Hsp100 conformations (New)

Figure 2—figure supplement 2: Functional ATPase modules are retained in the Hsp104 filament (former Figure 2—figure supplement 1). The figure was extended and includes now the superimposed L/S* modules from hexameric ScHsp104 on the AAA1 and AAA2 L/S* modules of the CtHsp104 filament.

Figure 2—figure supplement 3: Sequence conservation of HSP100 disaggregases (former Figure 2—figure supplement 2)

Figure 6—figure supplement 1: Position of the PS1-hairpin in hexameric Hsp104 (New)

Figure 6—figure supplement 2: Effect of PS1-hairpin deletion on ScHsp104 activity (former Figure 6—figure supplement 1)

Figure 6—figure supplement 3: SEC profiles of CtHsp104 dWA and dWB mutants (former Figure 6—figure supplement 2)

Figure 6—figure supplement 4: Effect of casein on ATPase activity (former Figure 6—figure supplement 3)

*Reviewer #3:*

*The manuscript by Heuck et al. describes the 3.7A resolution crystal structure of a fungal Hsp104 together with biochemical and cross-linking/mass spectrometry (XL-MS) data. The manuscript is well written and experimental results are convincing. Perhaps one of the most interesting finding is that the CCD via motif-2 contacts the AAA1 large subunit of the neighboring protomer providing the structural basis for a functional role of the CCD in nucleotide signaling between neighboring ATPase modules. The latter has largely been inferred but never been demonstrated for an Hsp100 chaperone. The role of the PS1 motif in coordinating the two ATPase rings is novel and supported by biochemical experiments. The hypothesis that the CCD belt immobilizes the entrapped AAA1 ATPase modules is novel and substantiated in part by the XL-MS analysis using BS3 crosslinker. The only concern with this approach is that, because of the homo-oligomeric structures, intra- vs. inter-domain crosslinks would be difficult to differentiate.*

*1) It is not entirely clear why heterotypic contacts cannot be resolved by the recent cryoEM analysis of a proposed helical Hsp104 assembly (Yokom et al., 2016). Although it remains uncertain whether a helical assembly is physiological, is the proposed role of the PS1 motif compatible with a helical assembly?*

Our original statement pointed mainly to the limited resolution of the cryoEM study (6.5A resolution of the unmasked map). However, we agree that the heterotypic contact of the CtHsp104 could, in principle, be also observed at that resolution. A further difference is that Yokom et al. used AMPPNP in their structural study, whereas the CtHsp104 crystal structure reflects the ADP bound state. It is thus unclear, whether technical or functional reasons account for the absence of the PS1 loop in the AAA1 active site of the ScHsp104 cryoEM structure. We thus followed the advice of the referee and adapted the indicated statement. Most importantly, the coupling function of the PS1 motif should be also possible within the helical hexamer, as discussed in our response to the Editorial Comments.

“[…]the mechanistic importance of heterotypic communication is less clear and its molecular underpinnings were not resolved in the recent cryoEM analysis (Yokom et al., 2016).”